# Neuronal pentraxin 2 is required for facilitating excitatory synaptic inputs onto spinal neurons involved in pruriceptive transmission in a model of chronic itch

Kensho Kanehisa[1,2], Keisuke Koga[1,3], Sho Maejima[4], Yuto Shiraishi[2], Konatsu Asai[2], Miho Shiratori-Hayashi[1,2], Mei-Fang Xiao[5], Hirotaka Sakamoto [4], Paul F. Worley[5] & Makoto Tsuda [1,2✉]

An excitatory neuron subset in the spinal dorsal horn (SDH) that expresses gastrin-releasing peptide receptors (GRPR) is critical for pruriceptive transmission. Here, we show that glutamatergic excitatory inputs onto GRPR$^+$ neurons are facilitated in mouse models of chronic itch. In these models, neuronal pentraxin 2 (NPTX2), an activity-dependent immediate early gene product, is upregulated in the dorsal root ganglion (DRG) neurons. Electron microscopy reveals that NPTX2 is present at presynaptic terminals connected onto postsynaptic GRPR$^+$ neurons. NPTX2-knockout prevents the facilitation of synaptic inputs to GRPR$^+$ neurons, and repetitive scratching behavior. DRG-specific NPTX2 expression rescues the impaired behavioral phenotype in NPTX2-knockout mice. Moreover, ectopic expression of a dominant-negative form of NPTX2 in DRG neurons reduces chronic itch-like behavior in mice. Our findings indicate that the upregulation of NPTX2 expression in DRG neurons contributes to the facilitation of glutamatergic inputs onto GRPR$^+$ neurons under chronic itch-like conditions, providing a potential therapeutic target.

[1] Department of Life Innovation, Graduate School of Pharmaceutical Sciences, Kyushu University, Fukuoka 812-8582, Japan. [2] Department of Molecular and System Pharmacology, Graduate School of Pharmaceutical Sciences, Kyushu University, Fukuoka 812-8582, Japan. [3] Department of Neurophysiology, Hyogo College of Medicine, Nishinomiya, Hyogo 663-8501, Japan. [4] Ushimado Marine Institute, Graduate School of Natural Science and Technology, Okayama University, 130-17 Kashino, Ushimado, Setouchi 701-4303, Japan. [5] The Solomon H. Snyder Department of Neuroscience, Johns Hopkins University School of Medicine, Baltimore, MD 21205, United States. ✉email: tsuda@phar.kyushu-u.ac.jp

tch is an unpleasant sensation that elicits an urge to scratch. Intrinsically, scratching behavior has a protective role in removing harmful substances such as chemicals and mites from the skin. However, in some pathological conditions, severe itching and repetitive scratching lead to worsening skin damage, inflammation, and itch sensitization (itch-scratch cycle). As the existing treatments (e.g., anti-histamines) are largely ineffective, elucidation of the mechanisms underlying chronic itch and the development of novel anti-itch therapies are major clinical challenges.

A growing body of literature has advanced our understanding of neural circuits involved in transmitting itch sensation from the skin to the brain[1,2]. Studies have shown that a subset of gastrin-releasing peptide receptor-expressing neurons (GRPR+ neurons) in the spinal dorsal horn (SDH) are indispensable for itch processing (evoked by chemicals), but are not involved in pain sensation[3–5]. Indeed, the ablation of GRPR+ neurons in mice caused a marked suppression of itch-related behavior (scratching) evoked by intradermal injection of pruritogens including histamine or chloroquine without affecting behavioral responses induced by nociceptive pain stimuli[4]. In animal models of chronic itch, GRP-GRPR signaling in the SDH is enhanced[6–8], and intrathecal injection of GRPR antagonists or ablation of GRPR+ SDH neurons suppresses scratching behavior[3,4]. Besides GRP, the principal fast neurotransmitter glutamate has also been reported to control the activity of GRPR+ neurons[9–11]. However, whether glutamatergic inputs onto GRPR+ neurons are altered in chronic itch models remains unknown.

In this study, we investigate the glutamatergic excitatory synaptic responses in SDH GRPR+ neurons using animal models of chronic itch related to atopic and contact dermatitis. Further, using several genetic tools we elucidate the role of neuronal pentraxin 2 (NPTX2; also known as NARP), an activity-dependent immediate early gene product[12,13], in the facilitation of excitatory synaptic responses in GRPR+ neurons in chronic itch models. Therefore, our findings represent a mechanism that could potentially be a target for treating chronic itch.

## Results

### Glutamatergic synaptic facilitation in GRPR+ SDH neurons in mouse models of chronic itch.

To investigate the excitatory synaptic responses in the SDH GRPR+ neurons, we first visualized these neurons using an adeno-associated viral (AAV) vector designed to express mCherry reporter protein under the control of mouse *Grpr* promoter (AAV-GrprP-mCherry). The AAV vector was microinjected into the cervical SDH of C57BL/6 mice (Fig. 1a) using a minimally invasive injection method[14]. In the AAV-GrprP-mCherry mice, cells expressing mCherry (mCherry+) were located in the superficial laminae of the SDH (Fig. 1b) and co-expressed NeuN (a neuronal marker) (Fig. 1c). RNAscope in situ hybridization confirmed that mCherry+ SDH neurons expressed *Grpr* mRNA (Fig. 1d). Consistent with previous data[15,16], a portion of mCherry+ neurons was positive for PAX2 (a marker of inhibitory interneurons; 176/902 total mCherry+ neurons tested, $n = 4$ mice) (Fig. 1e). Furthermore, whole-cell patch-clamp recordings of the mCherry+ neurons in the cervical spinal cord slices from AAV-GrprP-mCherry mice showed that injecting a depolarizing current evoked delayed firing (14/22 recorded cells; Fig. 1f) or transient firing pattern (2/22 recorded cells), both of which are known to occur in excitatory neurons. The percentage of mCherry+ neurons with each firing pattern was consistent with that of GRPR+ SDH neurons previously reported[8,16]. The resting membrane

potential (RMP) in mCherry+ neurons with delayed, transient, and tonic patterns was $-65.3 \pm 1.3$ mV, $-63.7 \pm 3.3$ mV, and $-64.4 \pm 2.3$ mV, respectively. There were no significant differences between the groups ($P = 0.942$, tonic vs. delay; $P = 0.985$, tonic vs. transient; $P = 0.908$, delay vs. transient). The RMP of SDH neurons with a delayed firing pattern was similar to the data in recent studies[8,17] but not in another study[16]. The reason for this difference remains unclear, but it may be related to several methodological differences (e.g., the age of the mice, preparation of spinal cord slices, and/or spinal segments). We also confirmed that following application of GRP (200 nM, a submaximal concentration to activate GRPRs[18]), the delayed firing mCherry+ SDH neurons depolarized and some of them fired action potentials (5/21 recorded cells) (Fig. 1g). The RMP of GRPR+ neurons with and without action potentials were $-64.4 \pm 1.8$ mV ($n = 5$) and $-63.5 \pm 1.8$ mV ($n = 16$), respectively. There was no significant difference between them ($P = 0.775$, unpaired $t$ test), indicating that the ability of GRP to induce action potentials is not dependent on the basal RMP. The GRP (300 nM)-evoked depolarization was inhibited in mCherry+ SDH neurons of GRPR knockout (KO) mice with intra-SDH injection of AAV-GrprP-mCherry (Post–Pre; wild-type (WT) mice, $8.55 \pm 1.25$ mV, $n = 7$ neurons; GRPR KO, $2.24 \pm 0.94$ mV, $n = 11$ neurons; $P = 0.0005$, Mann–Whitney test), confirming the expression of functional GRPRs in mCherry+ neurons. To verify the ability of mCherry+ neurons to produce itch-like behavior, we expressed the excitatory designer receptor hM3Dq by microinjecting AAV-GrprP-hM3Dq into the cervical SDH of WT mice. Systemic administration of clozapine-*N*-oxide (10 mg/kg, i.p.) markedly induced the scratching behavior in these mice (Fig. 1h). These results indicate that the AAV-GrprP-mCherry vector can be used to visualize the GRPR-expressing neurons involved in pruriceptive transmission in non-transgenic mice.

Using this tool, we next analyzed mCherry+ SDH neurons that displayed delayed firing patterns (hereafter referred to as GRPR+ neurons) and examined whether excitatory synaptic inputs to GRPR+ SDH neurons are changed under chronic itch-like conditions. We developed a mouse model of contact dermatitis by topical application of diphenylcyclopropenone (DCP)[7,8] to the nape of the neck of the AAV-GrprP-mCherry mice. First, we confirmed that the percentages of mCherry+ neurons with delayed, transient, and tonic firing patterns in DCP-treated mice were 70.4% (19/27 recorded neurons), 11.1% (3/27 recorded neurons), and 18.5% (5/27 recorded neurons), respectively, which was similar to that in control mice. The RMP in DCP-treated mice was slightly higher ($-63.2 \pm 1.7$ mV, $n = 16$ neurons) than that in vehicle-treated control mice ($-67.6 \pm 2.0$ mV, $n = 11$ neurons), but the difference was not statistically significant ($P = 0.134$, Mann–Whitney test). We found that spontaneous excitatory postsynaptic currents (sEPSCs) in GRPR+ SDH neurons were facilitated in DCP-treated mice compared to vehicle-treated control mice (Fig. 1i). The frequency and amplitude of sEPSCs in GRPR+ SDH neurons were significantly increased in DCP-treated mice (Fig. 1i–j). Similarly, both the frequency and amplitude of miniature EPSCs (mEPSCs) in GRPR+ neurons recorded in the presence of tetrodotoxin were also increased in DCP-treated mice (Supplementary Fig. 1a, b). Treating the slices with the α-amino-3-hydroxy-5-methyl-4-isoxazolepropionic acid receptor (AMPAR) antagonist, NBQX, abolished the sEPSCs in GRPR+ neurons in DCP-treated mice (Fig. 1k), indicating that synaptic facilitation to GRPR+ neurons is mediated by AMPARs. To test whether this synaptic facilitation to GRPR+ neurons is also observed in other models of chronic itch, we used the inbred strain NC/Nga mice, a model of atopic dermatitis, in which

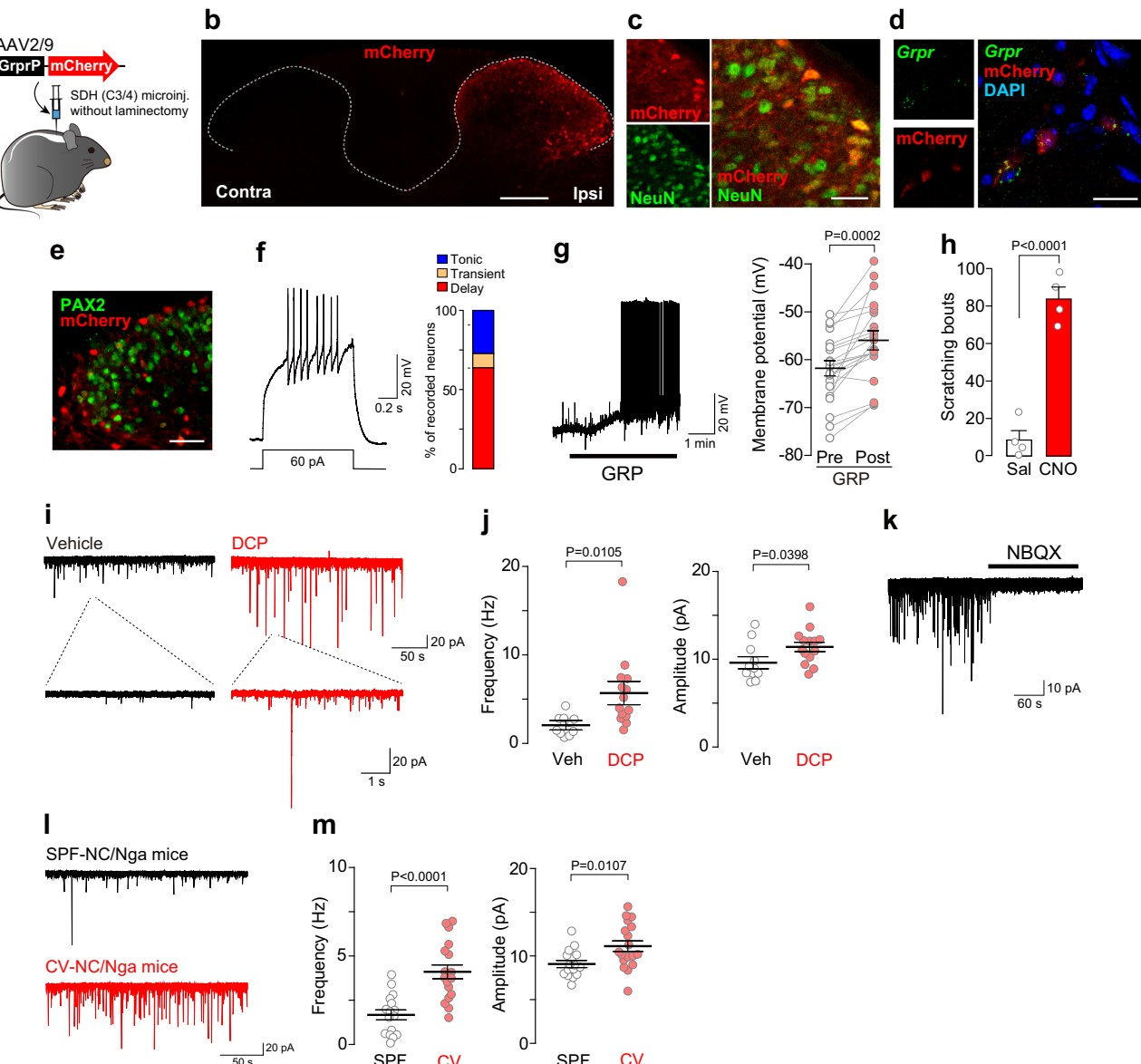

**Fig. 1 Facilitation of glutamatergic synaptic inputs onto GRPR⁺ SDH neurons in mouse models of chronic itch. a** Schematic illustration of microinjection of AAV-GrprP-mCherry into the cervical SDH. **b, c** Visualization of mCherry⁺ cells (red) in the SDH (**b**), and immunolabeling of mCherry⁺ cells (red) by a pan-neuronal marker, NeuN (green) (**c**). **d** RNAscope in situ hybridization for *Grpr* mRNA (green) in mCherry⁺ cells (red) in the SDH. **e** Immunolabeling of mCherry⁺ cells (red) by an inhibitory neuronal marker, PAX2 (green). Scale bars, 200 μm (**b**), 50 μm (**e**), 20 μm (**c, d**). **f** Firing patterns of mCherry⁺ neurons in cervical spinal cord slices. Representative traces of delayed firing pattern evoked by injecting a depolarizing current, and the percentage of mCherry⁺ neurons displaying each pattern ($n = 22$ cells). **g** Representative response of GRP (200 nM) in mCherry⁺ neurons (displaying the delayed firing pattern), and the summary of the membrane potential of pre- and post-GRP application ($n = 21$ cells, paired $t$ test). **h** Number of scratching behavior for 30 min after injection of clozapine-N-oxide (CNO, 10 mg/kg, i.p.) or saline in AAV-GrprP-hM3Dq mice ($n = 4$/group, unpaired $t$ test). **i, j** Representative traces (**i**) and the average of frequency and amplitude (**j**) of sEPSCs in GRPR⁺ (mCherry⁺) SDH neurons of vehicle- and DCP-treated mice (vehicle, $n = 11$ cells; DCP, $n = 14$ cells; unpaired $t$ test). **k** Representative traces of sEPSCs in GRPR⁺ (mCherry⁺) neurons of DCP-treated mice after application of NBQX, an antagonist for AMPARs. **l, m** Representative traces (**l**) and the frequency and amplitude (**m**) of sEPSC in SDH GRPR⁺ (mCherry⁺) neurons in SPF- and CV-NC/Nga mice (SPF, $n = 16$ cells; CV, $n = 18$ cells; unpaired $t$ test). Values represent mean ± S.E.M. Source data are provided as a Source Data file.

spontaneous scratching is observed when maintained under conventional (CV-NC/Nga), but not under specific-pathogen-free (SPF-NC/Nga) conditions[19]. Compared with SPF-NC/Nga mice, the frequency and amplitude of sEPSCs in GRPR⁺ SDH neurons were increased in CV-NC/Nga mice (Fig. 1l–m). These results provide evidence that AMPAR-mediated glutamatergic synaptic inputs onto GRPR⁺ neurons in the SDH are facilitated under chronic itch-like conditions.

**Toenail trimming prevents synaptic facilitation in GRPR⁺ SDH neurons.** Because the itch-scratch cycle is a critical component of chronic itch, we examined its involvement in the facilitation of synaptic responses in GRPR⁺ neurons by trimming the toenails of mice to prevent scratch-induced skin damage[7]. In DCP-treated mice we observed that trimming the toenail reduced the scratching behavior and skin lesion (Supplementary Fig. 2a, b), thereby suppressing the facilitation of synaptic responses in

GRPR+ neurons (Supplementary Fig. 2c, d). Thus, it is conceivable that the synaptic facilitation in GRPR+ neurons could be associated with signals from the inflamed itchy skin and/or repetitive scratching via primary afferents.

**Upregulation of NPTX2 in primary afferent neurons under chronic itch-like conditions**. We hypothesized that such signals could lead to an activity-dependent alteration in primary afferent sensory neurons, which are involved in facilitating excitatory synaptic responses in GRPR+ SDH neurons. To investigate this possibility, we focused on the role of NPTX2. NPTX2, also known as AMPAR-interacting partner, is an activity-dependent immediate early gene product in neurons, implicated in aggregating AMPARs[13,20] and strengthening excitatory synaptic transmission in the brain[21]. In the spinal cord neurons, NPTX2 has been reported to colocalize with AMPARs[20]. However, its role in synaptic transmission in the SDH and the chronic itch is unknown. We observed that *Nptx2* mRNA expression was upregulated in the DRG of DCP-treated mice compared to vehicle-treated control mice (Fig. 2a). The upregulation of *Nptx2* mRNA in the DRG was also observed in CV-NC/Nga mice (Fig. 2b). In contrast, the mRNA levels of *Nptx1*, another member of the neuronal pentraxin gene family[13], were not significantly different between the two groups (vehicle, $1.00 \pm 0.071$; DCP, $1.17 \pm 0.13$; $P = 0.310$), indicating a selective upregulation of *Nptx2* in the DRG under chronic itch-like conditions. The upregulation of *Nptx2* mRNA was significantly attenuated by trimming the toenails of the DCP-treated mice ($1.00 \pm 0.08$ and $0.72 \pm 0.08$ in DCP-treated mice without ($n = 8$ mice) and with toenail trimming ($n = 9$ mice), respectively; $P = 0.0074$, Unpaired $t$ test). Furthermore, immunohistochemical staining using a specific antibody for NPTX2[22] confirmed increased NPTX2 proteins in the DRG of DCP-treated mice (Fig. 2c). The observed NPTX2 immunostaining was specific since no staining was observed in NPTX2 KO mice (Supplementary Fig. 3a, b)[22]. In addition, NPTX2 KO did not affect the mRNA levels of *Nptx1* (a NPTX subtype) in the DRG (WT, $1.00 \pm 0.11$; KO, $0.88 \pm 0.09$; $n = 3$ mice). The percentage of NPTX2+ neurons per total DRG neurons and the intensity of NPTX2 immunofluorescence per DRG neuron were higher in DCP-treated mice than in vehicle-treated control mice (Fig. 2d). In contrast to the cervical segments, NPTX2 immunofluorescence in the lumbar segments (L4) of DCP-treated mice was very weak (Supplementary Fig. 3c), suggesting that NPTX2 upregulation could be restricted to the segments corresponding to the lesioned skin. In the cervical segment of DCP-treated mice, coimmunostaining revealed that most of the NPTX2+ DRG neurons co-expressed calcitonin gene-related peptide (CGRP) and tropomyosin receptor kinase A (TRKA), both of which are the markers of peptidergic neurons, and vesicular glutamate transporter 2 (VGLUT2) (Fig. 2e, f). A fraction of NPTX2+ DRG neurons co-expressed transient receptor potential vanilloid 1 (TRPV1). NPTX2-expressing DRG neurons were partially overlapped with neurons positive to neurofilament 200 (NF200) and, to a lesser extent, isolectin B4 (IB4) (Fig. 2e, f). Proteins synthesized in the DRG neurons are transported to their central terminals in the SDH. NPTX2 immunofluorescence was increased in the SDH of DCP-treated mice compared to the vehicle-treated control mice (Fig. 2g). Consistent with the colocalization of CGRP in DRG neurons, NPTX2 expression substantially overlapped with CGPR+ primary afferent terminals (Fig. 2g), implying that NPTX2 protein synthesized in the DRG neurons is transported to their central terminals in the SDH. This is supported by a previous study showing that NPTX2 immunofluorescence is markedly decreased by dorsal root rhizotomy[22]. We also observed much lower expression of

*Nptx2* mRNA in the spinal cord compared to the DRG in both vehicle- and DCP-treated mice by PCR (Fig. 2a) and in DCP mice by RNAscope in situ hybridization (Supplementary Fig. 3d). A similar pattern of NPTX2 upregulation in the DRG neurons, and its segment dependency and colocalization with CGPR+ DRG soma and SDH central terminals were also observed in the CV-NC/Nga mice (Supplementary Fig. 3e–h). Moreover, electron microscopy revealed that NPTX2 (small immunogold particles indicated by white arrowheads) was observed at a presynaptic terminal connected to a postsynaptic GRPR+ (mCherry+) neuron in the SDH (visualized by large immunogold particles indicated by red arrowheads) of DCP-treated mice (Fig. 2h). Together, these results suggest that under chronic itch-like conditions, NPTX2 is synthesized in the peptidergic DRG neurons and trafficked to their terminals in the SDH via primary afferent axons, and NPTX2+ terminals form synapses directly with GRPR+ SDH neurons.

**NPTX2 is essential for excitatory synaptic facilitation in GRPR+ SDH neurons in chronic itch model**. To determine the role of NPTX2 in the facilitation of excitatory synaptic inputs onto GRPR+ neurons, we employed NPTX2 KO mice. We found that the facilitation of sEPSCs observed in GRPR+ neurons of DCP-treated WT mice was suppressed in those of DCP-treated NPTX2 KO mice (Fig. 3a). Quantitatively, DCP-induced increase in the frequency and amplitude of sEPSCs were significantly lower in NPTX2 KO mice (Fig. 3b). Notably, both the sEPSC frequency and amplitude were indistinguishable between the vehicle-treated WT and NPTX2 KO mice (Fig. 3b). In addition, the percentage of GRPR+ neurons with the delayed firing pattern was similar between the WT and NPTX2 KO mice (WT, 60.9% (25/41 recorded neurons); NPTX2 KO, 70.0% (21/30 recorded neurons)). Thus, loss of NPTX2 suppressed the facilitation of glutamatergic synaptic responses in GRPR+ SDH neurons under chronic itch-like conditions without affecting basal excitatory synaptic transmission.

**Crucial role of NPTX2 expressed in primary afferent neurons in chronic itch-like behavior**. Finally, to determine whether NPTX2 plays a role in chronic itch, we observed the scratching behavior of DCP-treated NPTX2 KO mice. Compared with DCP-treated WT mice in which scratching behavior was markedly increased, NPTX2 KO mice displayed a significant reduction in DCP-induced scratching behavior (Fig. 4a). DCP-treated NPTX2 KO mice also had a lower dermatitis score (Fig. 4b) and transepidermal water loss (TEWL: an index of the skin barrier function) (Fig. 4c). In contrast, scratching responses induced by a single intradermal injection of chloroquine or compound 48/80, which are the models of histamine-independent and -dependent acute itch, respectively, were indistinguishable between the two genotypes (Fig. 4d). These results indicate that loss of NPTX2 attenuates the symptoms related to chronic itch without affecting acute itch-like behaviors. Since NPTX2 is also expressed in the brain[13], we next determined the role of NPTX2 specifically in DRG neurons. First, we examined whether the reduced scratching behavior in DCP-treated NPTX2 KO mice is rescued by overexpressing NPTX2 in the cervical DRG neurons. To achieve this, we locally injected AAV expressing NPTX2 (or GFP as a control) under the control of the neuronal promoter ESYN (AAV-ESYN-NPTX2 or -GFP) into the spinal nerves just distal to the DRGs at the cervical segments (C3 and 4). In mice injected with AAV-ESYN-GFP solution (containing blue dye), we observed that the DRG was dyed blue (Fig. 4e) and the DRG neurons expressed GFP (Fig. 4f), confirming the transgene expression in the cervical DRG neurons. To assess the scratching behavior in NPTX2 KO

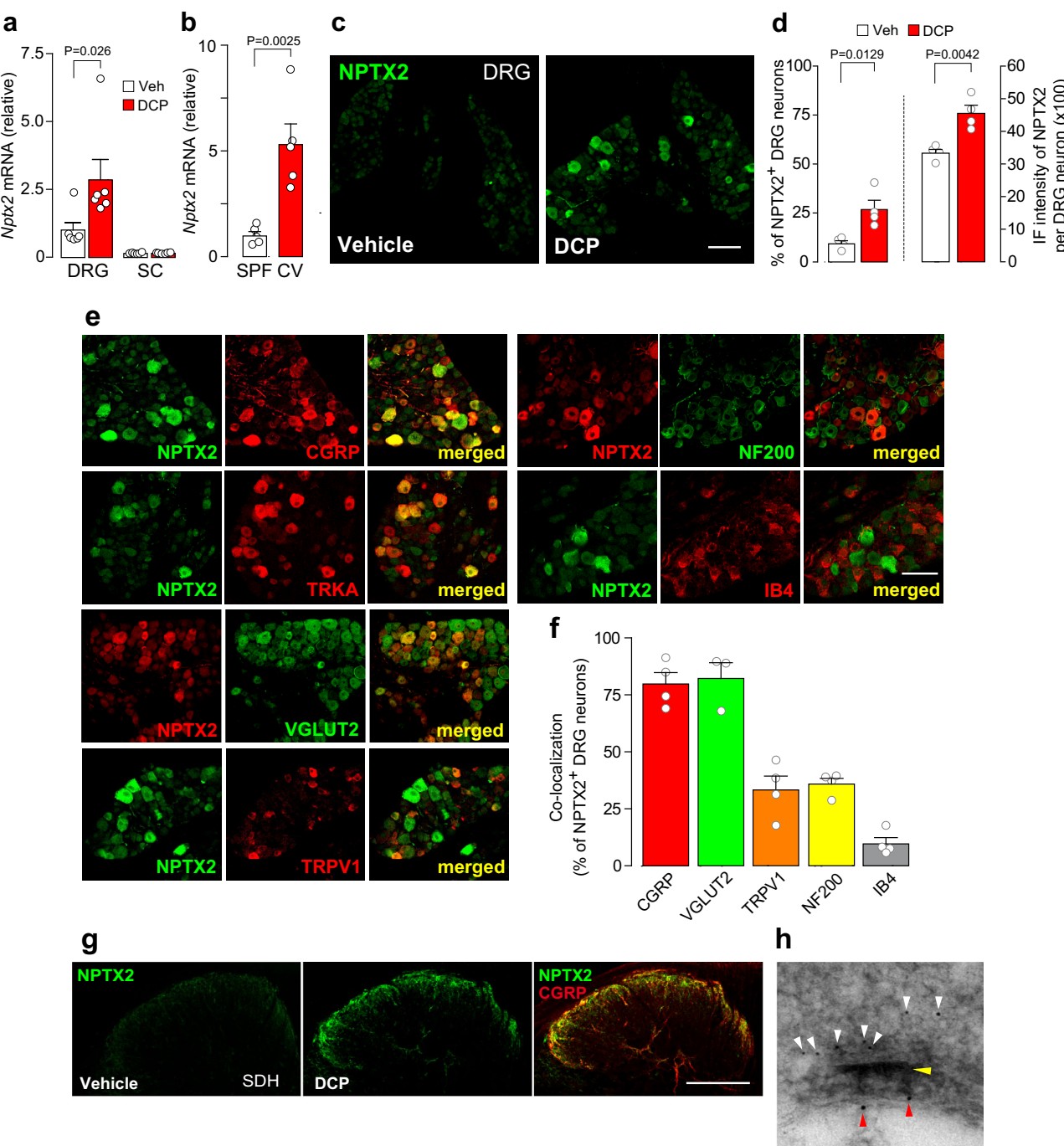

**Fig. 2 Upregulation of NPTX2 expression in peptidergic DRG neurons under chronic itch-like conditions. a**, **b** *Nptx2* mRNA in the cervical DRG and spinal cord (SC) in vehicle- and DCP-treated mice (**a**: $n = 6$/group, Mann–Whitney test) and in the cervical DRG in SPF- and CV-NC/Nga mice (**b**: $n = 5$/group, unpaired $t$ test). **c** NPTX2 immunofluorescence in the cervical DRG of vehicle- and DCP-treated mice. Scale bar, 100 μm. **d** Percentage of NPTX2$^+$ neurons in the cervical DRG and immunofluorescence (IF) intensity of NPTX2 per cervical DRG neurons in vehicle- and DCP-treated mice ($n = 4$/group, unpaired $t$ test). **e** Double-immunolabeling of NPTX2 and DRG neuronal markers (CGRP, TRKA, VGLUT2, TRPV1, NF200, and IB4) in the cervical DRG in DCP-treated mice. Scale bar: 100 μm. **f** Quantitative analysis of the percentage of NPTX2$^+$ neurons colocalized with DRG neuronal markers in DCP-treated mice (CGRP; $n = 4$, VGLUT2; $n = 3$, TRPV1; $n = 4$, NF200; $n = 4$, and IB4; $n = 4$). **g** NPTX2 immunofluorescence in the cervical SDH of vehicle- and DCP-treated mice. In DCP-treated mice, NPTX2 (green) was colocalized with CGRP (red) in the SDH. Scale bar, 200 μm.
**h** Electron microscopy analysis of NPTX2 expression at presynaptic terminal connected to a postsynaptic GRPR$^+$ (mCherry$^+$) neuron in the SDH. NPTX2 and mCherry were visualized by small (white arrowheads) and large (red arrowheads) immunogold particles conjugating antibody for NPTX2 and mCherry, respectively. Yellow arrowhead indicates the postsynaptic density site. Scale bar, 100 nm. Values represent mean ± S.E.M. Source data are provided as a Source Data file.

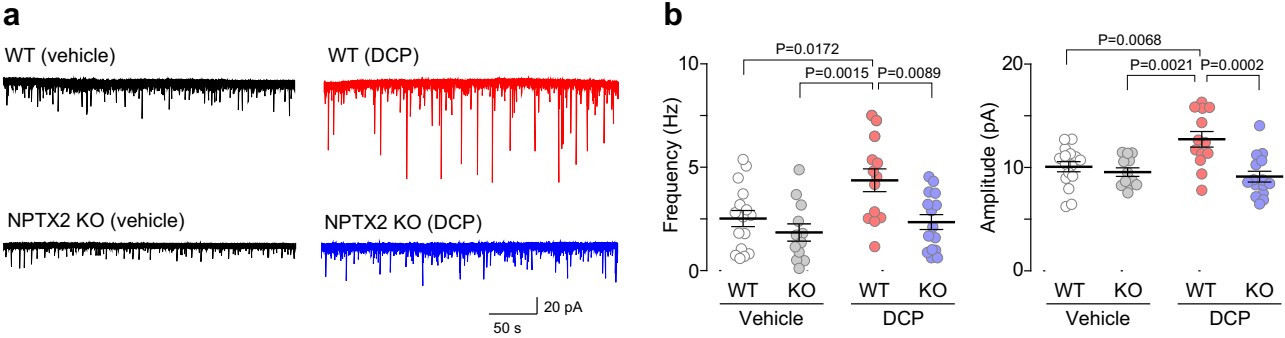

**Fig. 3 NPTX2 is indispensable for excitatory synaptic facilitation in GRPR⁺ SDH neurons in chronic itch model. a, b** Representative traces (**a**) and the frequency and amplitude (**b**) of sEPSCs in cervical spinal cord slices from vehicle- or DCP-treated WT and NPTX2 KO mice (vehicle: WT, $n = 16$ cells; KO, $n = 12$ cells; DCP: WT, $n = 13$ cells; KO, $n = 15$ cells; one-way ANOVA with post hoc Tukey's multiple comparisons test). Values represent mean ± S.E.M. Source data are provided as a Source Data file.

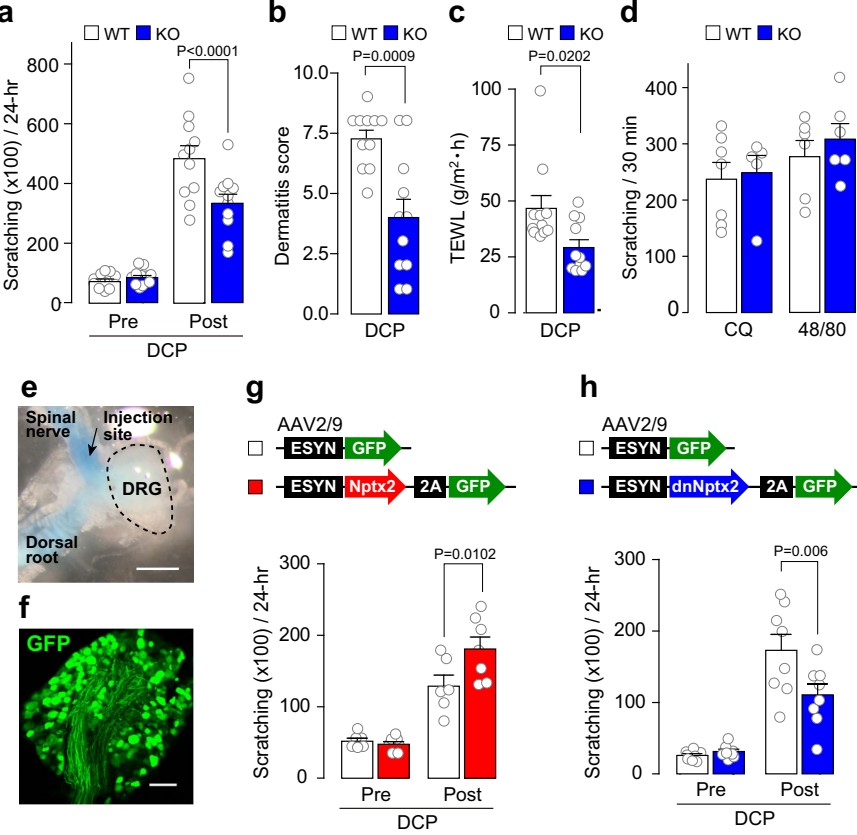

**Fig. 4 Role of NPTX2 in chronic itch-like behavior. a** Scratching behavior for 24 hr in WT and NPTX2 KO mice before (Pre) and after DCP treatment (Post) ($n = 11$ mice/group; two-way repeated-measures ANOVA with *post hoc* Bonferroni test). **b, c** Dermatitis score (**c**) and transepidermal water loss (TEWL) (**d**) in DCP-treated WT and NPTX2 KO mice ($n = 11$ mice/group). **d** Scratching bouts induced by intradermal injection of chloroquine (200 μg) and compound 48/80 (50 μg) in WT and NPTX2 KO mice (chloroquine: WT, $n = 7$; KO, $n = 5$; compound 48/80: WT, $n = 6$; KO, $n = 6$). **e** Photograph of the cervical DRG (C3) removed after the microinjection with AAV-ESYN-GFP including blue dye. Scale bar, 1 mm. **f** GFP fluorescence in the cervical DRG (C3) of mice with AAV-ESYN-GFP. Scale bar, 100 μm. **g** Scratching behavior by the left hind limb for 24 hr in NPTX2 KO mice with AAV-ESYN-GFP ($n = 6$) or AAV-ESYN-NPTX2 ($n = 7$) before (Pre) and after DCP treatment to the left back (Post). Two-way repeated-measures ANOVA with post hoc Bonferroni test. **h** Scratching behavior by the left hind limb for 24 hr in WT mice with AAV-ESYN-GFP ($n = 8$) or AAV-ESYN-dnNPTX2 ($n = 8$) before (Pre) and after DCP treatment to the left back (Post). Two-way repeated-measures ANOVA with post hoc Bonferroni test. Values represent mean ± S.E.M. Source data are provided as a Source Data file.

mice, we injected AAV-ESYN-NPTX2 or -GFP into the spinal nerves of the left C3/4 DRGs in order to minimize the time and tissue damage related to the operation, and DCP was topically applied only to the left side of the back and counted the number of scratching behaviors by the left hind limb. We found that the

reduction in scratching behavior in DCP-treated NPTX2 KO mice was rescued by ectopic expression of NPTX2 in cervical DRG neurons (Fig. 4g). Reciprocally, the DCP-induced scratching behavior in WT mice was significantly attenuated by expressing the dominant-negative form of NPTX2 (dnNPTX2), which

interferes with the secretion and function of endogenous NPTX2[20], in the left cervical DRG neurons (Fig. 4h). In addition, we confirmed that there was no significant difference in scratching behaviors between DCP-treated WT mice with and without the spinal nerve injection of AAV-ESYN-GFP, indicating that the surgery of the back skin and the injection of AAV vector themselves do not affect itch-related behavioral responses under our experimental conditions (Supplementary Fig. 4). These results suggest that NPTX2 expression in primary afferent sensory neurons critically contributes to chronic itch-like behavior.

## Discussion

Chronic itch is a clinically important symptom of many skin diseases; however, the underlying mechanism is poorly understood. Our study demonstrates the facilitation of AMPAR-mediated excitatory synaptic inputs onto GRPR[+] SDH neurons under chronic itch-like conditions and identifies a role of NPTX2 in regulating this process. Suppressing chronic itch-like behavior either by knocking out NPTX2 or by ectopically expressing dnNPTX2 in DRG neurons and rescue of the itch-related behavioral phenotype in NPTX2 KO mice by DRG neurons-specific ectopic expression of wild-type NPTX2 highlight the importance of DRG neuron-derived NPTX2. Given that NPTX2 expression was upregulated in DRG neurons under chronic itch-like conditions and that NPTX2 KO did not change the compound 48/80- and chloroquine-induced scratching (acute itch models), it appears that NPTX2 contributes to chronic itch. Although a possible involvement of NPTX2 in scratching behavior by other pruritogens cannot be ruled out, other signals evoked by neuropeptides (e.g., GRP) might be primarily involved in acute itch[5]. In addition, although the suppressing effect of toenail trimming on NPTX2 upregulation in the DRG was weaker than that on scratching behavior, the latter might also involve NPTX2-independent effects; for example, activation of astrocytes in the SDH (is crucial for chronic itch-like behavior and is suppressed by toenail trimming[7]) that was not affected by NPTX2 KO (Supplementary Fig. 5a).

NPTX2 expression in neurons is known to be activity-dependent[13]. In our study, NPTX2 upregulation in DRG neurons was selective to the segments corresponding to the inflamed itchy skin and was prevented by trimming the toenails. Thus, NPTX2 upregulation in DRG neurons could involve pruritic signals from the inflamed itchy skin and/or repetitive scratching. In the DRG, NPTX2 upregulation was mainly observed in CGRP[+] neurons, a population that also expresses VGLUT2[23–25]. Previous studies have shown that (1) CGRP, TRKA, and VGLUT2 are expressed in the DRG neuronal population related to pruriceptive transmission (NP2)[26], (2) CGRP[+] primary afferents elongate into the epidermis of the itchy skin[27] and contribute to pruriceptive transmission[28], (3) intrathecal administration of perampanel, a selective AMPAR antagonist, suppresses the scratching behavior of DCP-treated mice[29]. On the other hand, CGRP has been shown to be expressed in other populations of DRG neurons (e.g., PEP1 and 2)[26]. VGLUT2 is also broadly expressed in DRG neurons[26], and mice lacking VGLUT2 in primary afferent neurons exhibit spontaneous scratching behavior[24,30] (although this phenotype seems to be inconsistent between VGLUT2 KO mouse lines[24,25,30] and whether VGLUT2 in WT mice contributes to glutamatergic transmission from primary afferents to GRPR[+] SDH neurons[9,10] remains entirely unknown). If NPTX2 enhances nociceptive glutamate signaling, especially from VGLUT2[+] DRG neurons that can inhibit itch-like behavior and produce pain-like behavior, then DCP-treated mice in which NPTX2 is upregulated could display behavioral responses related to pain rather than itch. However, we found that mice treated with DCP in their

cheek (in which NPTX2 was also upregulated in trigeminal ganglion (TG) neurons) exhibited scratching but not wiping (Supplementary Fig. 6), the latter being an index of pain-like behavior[31]. In addition, although a putative role of NPTX2 expressed in SDH neurons in pain processing has also been considered[32], NPTX2 KO has been demonstrated to have no effect on behavioral responses associated with acute and chronic pain[22], and our study revealed only very faint signals of Nptx2 mRNA in the SDH of DCP-treated mice, which was in stark contrast to those in the DRG. Therefore, it is conceivable that under chronic itch-like conditions, NPTX2 expressed in primary afferent sensory neurons could preferentially strengthen glutamatergic synaptic inputs onto GRPR[+] neurons and is a critical factor in interlinking inflammatory itchy skin environment and excitatory synaptic facilitation in GRPR[+] SDH pruriceptive circuit, although the detailed mechanism needs to be investigated in the future. Nevertheless, we cannot exclude the possibility that under chronic itch-like conditions, NPTX2 is also upregulated in a nociceptor subpopulation (including VGLUT2[+] neurons) that can inhibit scratching behavior. If this occurs, why does NPTX2 KO suppress chronic itch-like behavior rather than enhance it? The underlying mechanism remains unclear, but given that glutamate released from VGLUT2[+] nociceptors might suppress itch-like responses by feedforward inhibition via inhibitory neurons in the SDH[33,34], a plausible explanation could be that the inhibitory SDH neurons cause dysfunction under chronic itch-like conditions, which results in reduced neuronal excitation by nociceptor-derived glutamate even if NPTX2 coexists. Indeed, a loss of feedforward inhibition by inhibitory SDH neurons in mice causes an excessive scratching phenotype[35,36], and nociceptive stimuli (e.g., scratching) fail to suppress itch in patients with atopic dermatitis[37]. Because it appears that the upregulation of NPTX2 expression occurs in several subpopulations of DRG neurons under chronic itch-like conditions, the identification of a subpopulation of NPTX2[+] DRG neurons that is responsible for chronic itch and other modalities, including pain, is an important subject in the future.

GRPR[+] SDH neurons have been shown to monosynaptically receive glutamatergic signals from primary afferent fibers[9,10]. A model described in brain synapses suggests that NPTX2 secreted from the presynaptic terminals aggregates AMPARs on the surface of the post-synapses[13,20]. This raises the hypothesis that NPTX2 synthesized in the DRG neurons is transported to the central terminals in the SDH and, after release, facilitates the glutamatergic synaptic transmission via clustering the AMPARs in GRPR[+] SDH neurons (Supplementary Fig. 7). This is consistent with the increase of sEPSC and mEPSC frequency and amplitude recorded from GRPR[+] SDH neurons in the condition related to chronic itch and is strongly supported by our findings that NPTX2[+] primary afferent terminals form synapses directly with GRPR[+] SDH neurons and that chronic itch-like behavior was suppressed by DRG neuron-specific ectopic expression of dnNPTX2 that interferes with the secretion of endogenous NPTX2[20]. While we emphasize the pivotal role of NPTX2 upregulated in DRG neurons in synaptic facilitation in GRPR[+] neurons and chronic itch-like behavior, it should also be noted that overexpression of NPTX2 in DRG neurons of WT mice did not induce spontaneous scratching. This could imply that NPTX2 expression alone is not sufficient to induce itch-like behavior and that additional signals, perhaps driving NPTX2 secretion from nerve terminals, may be needed. Furthermore, from our data showing that NPTX2 KO mice did not completely suppress DCP-induced scratching behavior, it is possible that other signals by neuropeptides (e.g., GRP)[5], may be involved in the residual scratching behavior in DCP-treated NPTX2 KO mice, which requires further investigation using tools (e.g., GRPR-specific

antagonists with a long-lasting effect or NPTX2/GRP double KO mice). Given that DCP-induced scratching seems to be histamine-dependent[38], neuromedin B, which is required for histaminergic itching[39] may also be involved. A recent study has demonstrated that GRPR+ SDH neurons depolarize progressively to a suprathreshold level by burst stimulation of GRP+ SDH interneurons and become excitable, and the suprathreshold excitation requires signals derived from GRP, but not from glutamate[16]. Thus, increased NPTX2 expression in DRG neurons seems to enhance the activity of GRPR+ SDH neurons under chronic itch-like conditions in which GRPRs are activated by GRP derived from primary afferents[3,5,40] and/or SDH interneurons[16] in response to pruriceptive signals from the skin (Supplementary Fig. 7). In future investigations, it is important to explore the contribution of GRP vs. glutamate to the activation of GRPR+ SDH neurons under chronic itch-like conditions.

Previously, we have shown that in animal models of chronic itch, GRP-induced excitation of GRPR+ SDH neurons is sensitized by lipocalin-2 (LCN2) derived from astrocytes in the SDH[7,8]. However, NPTX2 KO did not affect astrocyte activation and *Lcn2* mRNA expression (Supplementary Fig. 5a, b). Therefore, it is conceivable that under chronic itch-like conditions, NPTX2-mediated facilitation of glutamatergic excitatory synaptic responses and the astrocytic enhancement of GRP-mediated responses[7,8] concertedly render GRPR+ SDH neurons more excitable to pruriceptive signals from the inflamed skin, which is critical for chronic itch-like behavior.

In summary, we demonstrated the role of the activity-dependent gene product NPTX2 in the facilitation of excitatory synaptic inputs to GRPR+ SDH neurons and chronic itch-like behavior. Furthermore, NPTX2 upregulation, synaptic facilitation, and chronic itch-like behavior were attenuated by trimming the toenails. Thus, the vicious itch-scratch cycle, which is critical for chronic itch, seems to be a consequence of activity-dependent pathological alteration of the nervous system. Therefore, our findings represent a new mechanism that could be a target for specifically treating chronic itch.

## Methods

**Animals**. Male C57BL/6J mice were purchased from CLEA Japan (Tokyo, Japan). SPF and CV-NC/Nga mice (male, 10–15 weeks old) were purchased from SLC Japan (Shizuoka, Japan). Male NPTX2 KO mice (provided by professor Paul Worley) were used. All mice (except NC/Nga mice) used were 8–12 weeks of age at the start of each experiment and were housed at temperature and humidity ranges of 21–23 °C and 40–60%, respectively, with a 12-hr light–dark cycle. All animals were fed food and water ad libitum. All animals were housed in standard polycarbonate cages in groups of same-sex littermates. All animal experiments were conducted according to relevant national and international guidelines contained in the 'Act on Welfare and Management of Animals' (Ministry of Environment of Japan) and 'Regulation of Laboratory Animals' (Kyushu University) and under the protocols approved by the Institutional Animal Care and Use Committee review panels at Kyushu University.

**Recombinant adeno-associated virus (rAAV) vector production**. We constructed pZac2.1-GrprP-mCherry by cloning from a mouse extracted DNA of the spinal cord. To produce rAAV vector for *Grpr* promoter-dependent gene transduction, a vector containing the *Grpr* promoter (NCBI Reference Sequence: NM_000086.7; 1380 bp; −1352 to +28 [0 = transcription start site of the exon 1]) was generated from pZac2.1 by substituting the ESYN promoter with the *Grpr* promoter. We then cloned mCherry and hM3Dq (amplified from Addgene #45547) into the above-modified pZac2.1 to generate pZac2.1-GrprP-mCherry-WPRE and pZac2.1-GrprP-hM3Dq-WPRE, respectively. We cloned *Nptx2* gene using pCMV6-*Nptx2* (ORF) purchased from Origene (MR206833) into pZac2.1-ESYN promoter (pZac2.1-ESYN-Nptx2). The gene encoding the dominant-negative form of NPTX2 (dnNPTX2)[20] was provided by Prof. Paul Worley and was subcloned into pZac2.1-ESYN promoter (pZac2.1-ESYN-dnNPTX2). The rAAV vectors were produced from human embryonic kidney 293 T (HEK293T) cells with triple transfection [each pZac2.1 plasmid; pAAV2/9, trans plasmid; pAd DeltaF6, adenoviral helper plasmid (the latter two plasmids were purchased from the University of Pennsylvania Gene Therapy Program Vector Core)]. Viral lysate was harvested at 72 hr post-transfection and lysed by freeze-and-thaw cycles, purified

through two rounds of CsCl ultracentrifugation, and then concentrated using Vivaspin 20 ultrafiltration units (SARSTEDT, Germany). The genomic titer of rAAV was determined by Pico Green fluorometric reagent (Molecular Probes, USA) following denaturation of the AAV particle. Vectors were stored in aliquots at −80 °C until use.

**Microinjection of rAAV into the cervical SDH and DRG**. According to our previous study[41], mice were deeply anesthetized by s.c. injection of ketamine (100 mg/kg) and xylazine (10 mg/kg) and was shaved on the back of the neck. After the skin was incised, the muscle on C3–C5 vertebrae was opened with a retractor, and mice were attached with a head-holding device (SR-AR, NARISHIGE, Japan). Paraspinal muscles around the left side of the interspace between C3 and C4 vertebrae were removed, and the dura mater and the arachnoid membrane were carefully incised using the tip of a 30 G needle to make a small window to allow a glass microcapillary insert directly into the SDH. The glass microcapillary was inserted into the SDH (150–200 μm in depth from the surface of the dorsal root entry zone) with a pre-flow of rAAV solution through the small window (~500 μm lateral from the midline).

For microinjection into the cervical spinal nerves of the DRGs (C3 and C4), we carefully removed the muscle covering the cervical spinal nerves. The glass microcapillary was inserted directly into the spinal nerves just distal to the left cervical DRGs (C3 and C4). The unilateral injection (left side) was to minimize the time and tissue damage related to the operation.

rAAV solution was pressure-ejected (100 nL/min) for 5 and 3 min (~500 nL in SDH or 300 nL in spinal nerve, respectively) using the Micro Syringe Pumps (SYS-micro4, WPI, USA). After microinjection, the inserted glass microcapillary was removed from the SDH or spinal nerve, the skin was sutured with 3-0 silk, and mice were kept on a heating light until recovery. Three weeks later, these mice were used for all experiments.

**Immunohistochemistry**. As we previously reported[7], mice were deeply anesthetized by i.p. injection of pentobarbital (100 mg/kg) and perfused transcardially with 20 mL of phosphate-buffered saline (PBS; Wako, Japan), followed by 50 mL ice-cold 4% paraformaldehyde/PBS. The C3–C5 segments of the spinal cord, C3, C4, and L4 DRGs, or TG were removed, post-fixed in the same fixative for 3 hr at 4 °C, placed in 30% sucrose solution for 48 hr at 4 °C, and stored at −80 °C. Transverse spinal cord, DRG, and TG sections (30 μm) were incubated in blocking solution (3% normal goat or donkey serum) for 2 hr at room temperature and then incubated for 48 hr at 4 °C with primary antibodies: mouse anti-NeuN (1:2000; ab104224, Abcam, UK), rabbit anti-PAX2 (1:1000; 71-6000, Invitrogen), goat anti-CGRP (1:2000; ab36001, Abcam), rat anti-TRKA (1.500; AF1056, R&D Systems), guinea pig anti-TRPV1 (1:2000; GP14100, NEUROMICS), and biotinylated IB4 conjugates (1:2000; I21414, Invitrogen), chicken NF200 (1:1000; CH23015, NEU-ROMICS), guinea pig anti-VGLUT2 (1:500; Af810, Frontier Institute), rat anti-GFAP (1:2000; 13-0300, Invitrogen) and rabbit anti-NPTX2 (1:5000, provided by professor Paul Worley). After incubation, tissue sections were washed and incubated for 3 hr at room temperature in secondary antibody solution (Alexa Fluor 488, 546 and/or 405, 1:1000; Molecular Probes, USA) and streptavidin Alexa Fluor 405 (1:1000, S32351, Invitrogen). For Nissl staining, DRG sections were stained with Blue NeuroTrace Fluorescent Nissl Stains (in PBS, 1:100, N21479, Molecular Probes) for 20 min at room temperature after washing off the secondary antibodies. The tissue sections were washed, slide mounted, and subsequently coverslipped with Vectashield hardmount (Vector Laboratories, USA). Immunofluorescence images were obtained with a confocal laser microscope (LSM700, Carl Zeiss, Germany), in which the levels of NPTX2 immunofluorescence intensity of acquired images were not saturated. For quantitative analysis of NPTX2 immunofluorescence, we counted the number of NPTX2+ neurons in the DRG of control and DCP-treated mice and measured the immunofluorescence intensity of NPTX2 on a cell-by-cell basis. Under our criterion that DRG neurons in which the S/N ratio of NPTX2 immunofluorescence was ≥4.0 were counted as being positive for NPTX2.

**RNAscope in situ hybridization**. Mice were deeply anesthetized by i.p. injection of pentobarbital and perfused transcardially with ice-cold PBS followed by 50 mL ice-cold 4% paraformaldehyde/PBS. The C3 to C5 spinal cord and C3 DRG were quickly removed, post-fixed in the same fixative for 3 hr at 4 °C, placed in 30% sucrose solution for 48 hr at 4 °C, and stored at −80 °C until use. Tissues were embedded in O.C.T compound (Sakura Finetek Japan, Tokyo, Japan) and made at a slice thickness of 14 μm. Fluorescent in situ hybridization (ACDbio, CA, USA) was performed following the manufacturer's instructions for frozen tissue. Using probes were listed below. Probes: Mm-Grpr (ACDbio, 317871, CA, USA), Mm-Nptx2 (ACDbio, 316901, CA, USA). Tissue sections were analyzed using an LSM700 Imaging System (Carl Zeiss, Oberkochen, Germany).

**Slice preparation and electrophysiology**. According to our previous study[8], mice were deeply anesthetized with urethane (1.2–1.5 g/kg), and the cervical spinal cord was removed and placed into a cold high sucrose artificial cerebrospinal fluid (sucrose aCSF) (250 mM sucrose, 2.5 mM KCl, 2 mM CaCl₂, 2 mM MgCl₂, 1.2 mM NaH₂PO₄, 25 mM NaHCO₃, and 11 mM glucose). A parasagittal spinal cord slice

(250–300 μm thick) was made using a vibrating microtome (VT1200, Leica, Germany) and then the slices kept in oxygenated aCSF solution (125 mM NaCl, 2.5 mM KCl, 2 mM CaCl$_2$, 1 mM MgCl$_2$, 1.25 mM NaH$_2$PO$_4$, 26 mM NaHCO$_3$, and 20 mM glucose) at room temperature (22–25 °C) for at least 30 min. The spinal cord slice was then put into a recording chamber where it was continuously superfused with aCSF solution at 25–28 °C at a flow rate of 4–6 mL/min. The patch pipettes were filled with an internal solution (125 mM K-gluconate, 10 mM KCl, 0.5 mM EGTA, 10 mM HEPES, 4 mM ATP-Mg, 0.3 mM NaGTP, 10 mM phosphocreatine, pH 7.28 adjusted with KOH), and whole-cell patch-clamp recordings were made from mCherry$^+$ SDH neurons. Recordings were made using Axopatch 700B amplifier and pCLAMP 10.4 acquisition software (Molecular Devices, USA). The data were digitized with an analog-to-digital converter (Digidata 1550, Molecular Devices), stored on a personal computer using a data acquisition program (ClampeX version 10.4, Molecular Devices). The drugs used were GRP (200 or 300 nM; 4011671, Bachem, Switzerland) and NBQX (10 μM in 0.1% DMSO, 14914, Cayman, USA). The firing patterns of mCherry$^+$ neurons were determined in a current-clamp mode by passing depolarizing current pulses (60 pA) for 1 s through the recording electrode from the resting membrane potential[8]. Spontaneous EPSCs (sEPSCs) were recorded for 10 min under holding the membrane potential at −70 mV in a voltage-clamp mode. Miniature EPSCs (mEPSCs) were recorded 10 min after the treatment of spinal cord slices with tetrodotoxin (1 μM). The frequency and amplitude of sEPSCs and mEPSCs for 10 min were averaged using the Mini Analysis Program (Synaptosoft).

**Mouse models of acute and chronic itch.** For acute itch models, mice were shaved on the back until one day before injection. Intradermal injection of pruritogens [chloroquine (200 μg/50 μL; C6628, Sigma) and compound 48/80 (50 μg/50 μL; C2313, Sigma)] into the shaved back was performed, and measured scratching behavior for 30 min as described below.

To induce contact dermatitis, mice were shaved on the back and topically applied by painting 0.2 mL of 1–2% DCP (046-26741, Wako)[42] dissolved in acetone under isoflurane anesthesia[7]. For the experiments in Fig. 4g, h, DCP was topically applied only to the left side of the back because we injected AAV vectors into the left spinal nerve only. For the experiment in Supplementary Fig. 6, mice were shaved on the cheek and topically applied with 0.05 mL of 2% DCP. Seven days after the first painting (day 7), DCP was painted again on the same area of skin. Seven days later (day 14), scratching behavior and other experiments for whole-cell recordings (day 13 or 14), immunohistochemistry, real-time PCR, dermatitis score, transepidermal water loss were performed. For the experiments in Fig. 4g, h, the number of scratching behaviors by the left hind limb were counted.

For a chronic itch model associated with atopic dermatitis, NC/Nga mice (10–15-week-old) that were maintained under CV conditions were used[7,19]. As a control, NC/Nga mice (10–15-week-old) housed in specific-pathogen-free (SPF) conditions were used.

**Trimming of toenails.** As described in our previous study[7], to prevent skin damage by scratching, all toenails of the bilateral hindpaws of mice were trimmed every three or four days under isoflurane anesthesia (Supplementary Fig. 2a).

**Real-time reverse-transcription polymerase chain reaction.** As we previously reported[7], mice were anesthetized with pentobarbital and perfused transcardially with PBS. The C3–C5 segments of the spinal cord and DRG or the L3–L5 segments of the DRG were removed immediately. Total RNA was extracted using TRIsure (Bioline, UK) according to the manufacturer's protocol. The amount of total RNA was quantified by measuring the optical density at 260 nm (OD260) using a spectrophotometer (Nanodrop One, Thermo Fisher, USA). For reverse transcription, 250 ng of total RNA was transferred to the reaction with Prime Script reverse transcriptase (Takara, Japan) and random 6-mer primers. Quantitative polymerase chain reaction (PCR) was carried out with Fast Start Essential DNA Probes Master or Fast Start Essential DNA Green Master (Roche, Switzerland) using LightCycler 96 (Roche) according to the manufacturer's specifications, and the data were analyzed by LightCycler 96 Software (Roche) using standard curves. Values were normalized to the level of *Gapdh* mRNA. The TaqMan probe, forward primer, and reverse primer used in this study were as follows: *Gapdh*, probe, 5′-FAM-ACCAC CAACTGCTTAGCCCCCCTG-TAMRA-3′; forward primer, 5′- GCCCCCATG TTTGTGATG-3′; reverse primer, 5′-GGCATGGACTGTGGTCATGA-3′. The primers and probe for *Nptx1* (Mm.PT.58.6801270) and *Nptx2* (Mm.PT.58. 31290939) were obtained from Integrated DNA Technologies (IA, USA). The primers and probe for *Lcn2* (Mm01324470_m1, Thermo Fisher Scientific) were used.

**Measurement of scratching behavior.** Mice were placed individually in a plastic chamber (11 cm in diameter, 18 cm high), and habituated for 0.5–1 hr to allow acclimatization to the new environment. In acute itch model, hind limb scratching behavior directed toward the injection site was videotaped for 30 min after intradermal injection of pruritogens. According to our previous study[7], one scratch bout was defined as a lifting of the hind limb toward the injection site and then placing the limb back on the floor. In the cheek model, the number of bouts of scratching with the hindpaw and wiping with the forepaw was counted. In chronic itch

models, scratching behavior was automatically detected and objectively evaluated using MicroAct (Neuroscience, Japan) in accordance with a method described previously[7]. Briefly, under isoflurane anesthesia, a small Teflon-coated magnet (1 mm in diameter, 3 mm in length, Neuroscience) was implanted subcutaneously into the hindpaws of the mice at least 1 day before the first measurement of scratching behavior (for the experiments in Fig. 4g, h, the magnet was implanted into the left hindpaw only). Each mouse with implanted magnet was placed in an observation chamber (11 cm in diameter, 18 cm high) with food and tap water, surrounded by a round coil. The movement of magnets implanted subcutaneously into the hindpaws induced electric currents in the coil, which were amplified and recorded by MicroAct software. The analysis parameters for detecting scratch movements were: threshold, 0.07 V; event gap, 0.2 s; minimum duration, 0.2 s; maximum frequency, 35 Hz; minimum frequency, 2 Hz; minimum beats, 2. Scratching behavior was shown as the number of total scratching strokes over 24 hr.

**Post-embedding immunoelectron microscopy.** Mice injected with AAV-GrprP-mCherry 14 days after DCP treatment were anesthetized and perfused transcardially with 4% paraformaldehyde and 0.1% glutaraldehyde/PBS. Spinal cords were immediately removed and immersed in 4% paraformaldehyde/PBS for 3 hr at 4 °C. Then, cervical cords (C3/4) were sectioned in the transverse plane at 200 μm in thickness with a Linear Slicer (PRO10, Dosaka EM, Japan). Preparations were dehydrated through increasing concentrations of methanol, embedded in LR Gold resin (Electron Microscopy Sciences, USA), and polymerized under UV lamps at –20 °C for 24 hr. Ultrathin sections (~70 nm in thickness) were collected on nickel grids coated with a collodion film, rinsed with PBS several times, then incubated with 2% normal goat serum and 2% BSA in 50 mM Tris(hydroxymethyl)-amino-methane-buffered saline (TBS; pH 8.2) for 30 min to block non-specific binding. The sections were then incubated with a mixture of rabbit anti-NPTX2 antibody (1:50) and chicken anti-mCherry antibody (1:400; ab205402, Abcam) for 1 hr at room temperature to visualize the mCherry signals under the electron microscope. To intensify the immunoreactivity for NPTX2, a streptavidin-biotin intensification kit (Nichirei, Japan) was used, and first incubated with the biotinylated goat anti-rabbit IgG antibody for 10 min at room temperature, followed by incubation in the avidin-biotin-horse radish peroxidase (HRP) complex solution for 5 min at room temperature. The sections were then washed with PBS, incubated with a mixture of goat antibody against chicken IgY conjugated to 10 nm gold particles (1:50; ab41511, Abcam) and goat antibodies against HRP conjugated to 6 nm gold particles (1:50; 145687, Jackson ImmunoResearch Laboratory, USA) for 1 hr at room temperature. Finally, the sections were contrasted with uranyl acetate and lead citrate and viewed using an H-7650 (Hitachi, Japan) electron microscope operated at 80 kV.

**Evaluation of dermatitis.** Severity of dermatitis of the face, ears, and the rostral part of the body was assessed as previously described[7,19], no symptoms (score 0), mild (score 1), moderate (score 2), and severe (score 3). This scoring system was separately applied to the severity of erythema/hemorrhage, edema, excoriation/erosion, and scaling/dryness[43,44]. The total score (minimum 0, maximum 12) was expressed as the sum of each score of the above four symptoms. The detailed description of the scoring criteria for dermatitis is as follows: erythema/hemorrhage of the rostral back skin, 0 (no erythema/hemorrhage), 1 (local erythema, no hemorrhage on the rostral back skin), 2 (disseminated erythema, no hemorrhage), 3 (erythema on the entire rostral back skin or hemorrhage caused by repeated scratching); edema in the ear pinna, 0 (no increase in ear thickness), 1 (slight increase in thickness in either the left or right ear pinna), 2 (marked increase in thickness of both sides of ear pinna), 3 (marked increase in thickness and stiffness of both sides of ear pinna); excoriation/erosion in the ear pinna, 0 (no excoriation and tissue deficit), 1 (local (not continuous) excoriation, no tissue deficit), 2 (small scale continuous excoriation, no tissue deficit), 3 (continuous excoriation and tissue deficit); scaling/dryness of the rostral back skin, 0 (no scaling/dryness), 1 (local scaling and slight exfoliation of skin), 2 (disseminated scaling or marked exfoliation of skin), 3 (scaling of the entire area and marked exfoliation of skin).

**Measurement of transepidermal water loss.** TEWL was measured using the Tewameter TM300 system and a multi-probe adaptor (CK electronic, USA), in accordance with manufacturer instructions and our previous study[8]. Under isoflurane anesthesia, the probe collar was placed on the surface of the skin on the animal's back for 20–30 s. Measurements were obtained twice for the left and right sides of the skin, and the values were averaged.

**Statistics and reproducibility.** All data are shown as the mean ± SEM. Statistical significance of differences was determined using paired *t* test (Fig. 1g), unpaired *t* test (Fig. 1h, j, m, 2b, d, 4b, d (48/80), and Supplementary Figs. 1b left, 2b, 2d left, and 5b), Mann–Whitney test (Figs. 2a, 4d, d (CQ), and Supplementary Figs. 1b right, 2d right, 4, 6a), one-way analysis of variance (ANOVA) with post hoc Tukey's multiple comparisons test (Fig. 3b), two-way repeated-measures ANOVA with post hoc Bonferroni test (Fig. 4a, g, h), by using GraphPad Prism 4 and 7 software (GraphPad Software, USA). Differences were considered significant at a *P* value of <0.05. Confocal or EM micrographs shown in Figs. 1b–e, 2c, e, g, h, 4f,

and Supplementary Figs. 3a–h, 5a, and 6b were representative of three to five independent experiments. The photograph in Fig. 4f was representative of two independent experiments.

**Reporting summary**. Further information on research design is available in the Nature Research Reporting Summary linked to this article.

## Data availability

Data underlying the findings of the study are available from the corresponding author upon request. Source data are provided in this paper.

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

## Acknowledgements

This work was supported by JSPS KAKENHI Grant Numbers JP19K22500, JP19H05658, JP20H05900 (M.T.), JP16H06280 (H.S.), by NINDS R35 NS097966 (M.X. and P.F.W.), by Grant-in-Aid for Scientific Research on Innovative Areas—Platforms for Advanced Technologies and Research Resources "Advanced Bioimaging Support (ABiS)" (H.S.), by the Practical Research Project for Allergic Diseases and Immunology (Research on Allergic Diseases and Immunology) from AMED under Grant Number JP18ek0410034 (M.T.), by the Core Research for Evolutional Science and Technology (CREST) program from AMED under Grant Number JP21gm0910006 (M.T.), by Naito Foundation (M.T.) and by Platform Project for Supporting Drug Discovery and Life Science Research (Basis for Supporting Innovative Drug Discovery and Life Science Research (BINDS)) from AMED under Grant Number JP21am0101091 (M.T.).

## Author contributions

K.K. designed experiments, performed almost all experiments, analyzed the data, and wrote the manuscript. K.K., Y.S., K.A., M.S.-H. assisted some experiments. S.M. and H.S. performed electron microscopy experiments. M.X. and P.F.W. provided critical materials and advice on data interpretation. M.T. conceived this project, supervised the overall project, designed experiments, and wrote the manuscript.

## Competing interests

The authors declare no competing interests.

**Additional information**

