## [Peer Review File · Nature Communications]

Neuronal pentraxin 2 is required for facilitating excitatory synaptic inputs onto spinal neurons involved in pruriceptive transmission in a model of chronic itchREVIEWER COMMENTS

Reviewer #1 (Remarks to the Author):

Glutamatergic mechanisms regulating chronic itch are poorly understood. In this manuscript, Kanehisa et. al. demonstrate an essential role of NPTX2 in murine sensory neurons for facilitating excitatory synaptic inputs onto spinal GRPR neurons in two chronic itch models. Their electrophysiological recordings of AAV-GrprP-mCherry-infected spinal dorsal horn neurons indicate that AMPAR-mediated sEPSCs in GRPR neurons were significantly facilitated in mice with chronic itch. Interestingly, NPTX2, an AMPAR-interacting partner, is significantly up-regulated in DRG neurons under chronic itch conditions. Next, they show that NPTX2 is essential for excitatory synaptic facilitation in GrprP-mCherry neurons using loss of function and gain of function approaches. Overall, these findings are an interesting and important. However, major revision is necessary to improve the manuscript.

Major concerns

1. The authors should provide the counting data and the percentage of overlaps between NPTX2 and other markers (e.g. CGRP, IB4, TRPV1, NF200). Is GRP expressed in NPTX2 neurons? If so, do GRP/glutamate co-released in these chronic itch models. What is the role of GRP in glutamatergic facilitation of GrprP-mCherry neurons? This should be examined and compared between the control and mice in these pathological conditions.

2. Although AMPAR antagonist abolished the sEPSCs in GrprP-mCherry neurons in DCP-treated mice, and NPTX2 is known as AMPAR-interacting partner, it only implies the possible connection between NPTX2 fibers and GrprP-mCherry neurons. The role of glutamate in itch is unclear. Peripheral Vglut2-mediated glutamatergic transmission is not required for acute itch (Liu, et al; Neuron, 2010). Is Vglut2 expressed in NPTX2 neurons? Others showed the involvement of glutamate within the dorsal horn microcircuits (Wan et al, 2017, Aresh, et al, 2017). Thus, some evidence supporting the direct synaptic connection between NARP fibers and GrprP-mCherry neurons would be necessary to show whether the effect is direct or indirect.

3. Fig. 1d: what is the mean value of the resting potential? Did the authors find a correlation between the resting potential and the firing pattern of GrprP-mCherry neurons? The delayed pattern is more common between -70 and -80 mV. Several studies have characterized the electric properties of GRPR neurons, those studies should be cited and discussed. Fig. 1e: was the ability of GRP to induce action potentials dependent on neuron resting potential?

4. It is quite surprising that the facilitation of both spontaneous and miniature EPSCs consists only in an increase of frequency, with no effects on amplitude. If an increase of clustering of postsynaptic AMPA receptors is involved as the authors implied, I would expect also an increase of amplitude, unless only new AMPA-mediated synapses are involved and no changes in the number of AMPA receptors at the existing synapses occur. Are the cumulative distributions of miniature EPSC amplitudes similar in control and in mice with chronic itch.

5. Fig. 3 shows the critical evidence that chronic itch enhanced the expression of NPTX2 mRNA and protein in cervical DRG neurons. There are many weakly stained neurons in the control DRG. Considering the absence of staining in NPTX2 KO DRG (Sup. Fig. 2a), the weak staining in control DRG is probably real. Does DCP treatment increase the number of positive neurons, the signal intensities, or both? As mentioned in #1, all the immunostaining results should be quantified and compared.

Minor points

1. The durations of EPSC recording are very short (3 min for sEPSCs and 2 min for mEPSCs). These recordings usually show some variability and a longer duration (at least 10 min) is preferable.

2. Toenail trimming is interesting. How many times was the experiment repeated? What's the n number? The change is minimal comparing to the >2.5 fold increase of NPTX2 mRNA after DCP

painting. It's not even close to be prevented by toenail trimming as the authors stated in the discussion.

3. The scratching numbers vary greatly in different experiments. Control mice in Fig. 5 h and i scratched only about 150 times, which are much fewer than that in Fig. 5a and 2b. Is it caused by different painting procedures?

4. Supplementary Fig. 2e shows immunostaining images in the spinal cord of CV-NC/Nga mice. Were the mice painted with DCP as described in the figure legend?

5. The authors show that the two different chronic itch models are consistent in molecular changes. However, the two models are not identical. Have the authors examined the expression levels of NPTX2 in the lumbar DRGs of CV-NC/Nga mice?

6. What is the rationale for the dose of GRP used ?

7. Scale bars should be double checked.

Reviewer #2 (Remarks to the Author):

This is an interesting report on the role of nptx2 facilitating synaptic transmission between primary afferents and GRP+ spinal dorsal horn neurons in chronic itch conditions. The authors convincingly show that nptx2 is upregulated in primary afferents in two models of chronic itch and that spinal dorsal horn GRP+ neurons investigated in a slice preparation are rendered hyperexcitable under these conditions. Nptx2 was found upregulated in peptidergic primary afferents in the two models and genetic ablation of Nptx2 abolished hyperexcitability and reduced scratching behavior, which was rescued by re-expression of Nptx2 in primary afferents. Also, toenail trimming that reduced scratching-related skin damage including local inflammation and scratching behavior normalized neuronal excitability of spinal GRP+ neurons.

The authors suggest that Nptx2 activity dependently released from primary afferents innervating the inflamed skin facilitates spinal processing of itch.

Comments:

This is an important contribution in the field of interaction between peripheral input and spinal processing of chronic itch.

When considering the electrophysiological data one would love to see the characteristics of GRP neurons in the itch models (distribution of tonic, transient and delayed firing; membrane potential) – assuming that the chronic itch condition with ongoing peripheral input and GRP release modified the characteristics. If this is the case, the authors need to discuss how this might affect their findings of reduced sEPSP and mEPSP.

Upregulation of Nptx2 in primary afferents was quite broad suggesting that it might not have an effect restricted to pruriceptors, but also involves nociceptors. The authors have included a chronic inflammatory model – however, they tested 14 days after CFA – this appears somewhat late considering the time course of hyperalgesia with behavioral changes being maximum within a few days.

Moreover, there are reports on a potential role of Nptx2 in chronic pain

<https://doi.org/10.1101/2020.03.19.996645>; <https://doi.org/10.1016/j.cophys.2019.10.002>

that should be discussed. The comparison might reveal that rather than the difference between itch and pain there may be a crucial difference between acute and chronic sensitization. This would shed some light on the general problem we are facing when acutely evoked pain or itch models are used to investigate chronic conditions that are dominated by spontaneous itch/pain.

Reviewer #3 (Remarks to the Author):

The manuscript by Kanehisa et al. entitled “Neuronal pentraxin 2 is required for facilitating excitatory synaptic inputs onto spinal itch-transmission neurons under chronic itch conditions” describes work where the contribution of the pentraxin 2 protein to itch responses was examined. Pentraxin 1 and 2 have been previously reported to mediate the uptake of synaptic macromolecules and serve functions in synaptic plasticity. In particular, pentraxin 2 (Narp or NP2 or Nptx2) has been described as an activity-regulated protein which is secreted forming clusters with itself that co-aggregate with the AMPA receptor (O'Brien 1999 Neuron). To understand the potential mechanism, in the spinal cord, underlying chronic itch, the activity of GRPR neurons in the spinal cord was examined. As expected, activity was increased in the two models of chronic itch used. Global knockout of Nptx2 attenuated GRPR activity, moderately reduced itch behavioral responses, and improved skin condition in the DCP model of chronic itch. Rescue experiments where Nptx2 was expressed in DRG neurons in Nptx2 null mice are reported to recapitulate the increased itch sensitization to DCP treatment seen in wild-type mice.

The evidence provided for the proposed mechanism for Nptx2 mediate sensitization is insufficient with the tools used in the study inadequately characterized (point 1), unsatisfactory characterization of DRG neurons expressing described (point 5 & 6), and the design of experiment extremely poor and consequently of little use.

Major Problems

- 1) The study uses of an undefined DNA element upstream of the GRPR gene to drive expression of fluorescent markers and of a DREADDq chemogenetic receptor in spinal cord neurons. This method for identifying and examining the function of GRPR is not adequately tested. There needs to be experiments performed to directly demonstrate that GRPR-neurons are targeted, for instance, by ISH should be used to examine the expression of transgene versus GRPR. Also, the numbers and type of neurons reported here are at odds with those reported by Pagani et al, (2019 Neuron), there were inhibitory as well as excitatory neurons described in the latter paper.
- 2) CNQX is a potent AMPA/kainite receptor antagonist but is also an antagonist at the glycine modulatory site on the NMDA receptor complex. A comment needs to be added to address this as glycine release from inhibitory IN might be also effect with CNQX.
- 3) There is considerable controversy about whether primary afferents synapse directly on GRPR-neurons. The paper assumes the latter but provides not direct evidence for this interaction. At a minimum this point should be addressed in a fair way in the Discussion.
- 4) Figure 2 presents data that could easily be combined with the data in figure 1 or move to the supplements.
- 5) The level of Nptx2 is shown to be increased in DRG neurons in Figure 3. Nptx2 is expressed in many subtypes of DRG neurons, see Sharma et al., 2020 Nature https://kleintools.hms.harvard.edu/tools/springViewer_1_6_dev.html?datasets/Sharma2019/alland Usoskin et al, Nature Neurosci 2015; <http://linnarssonlab.org/drg/> Given these published results, why, under normal conditions, is there no detectable Nptx2 expression in DRG under naïve conditions?
- 6) In addition (Figure 3), the colocalization studies were performed with markers that have little relevance to itch. Recent work demonstrate that the NP2 and NP3 populations of neurons are responsible for itch. CGRP, TrkA, Trpv1, and IB4 are not selective for these populations of neurons. This cast serious doubts about whether Nptx2 is overexpressed in the appropriate classes of neurons. Indeed, a result of the reported chronic itch are skin lesions which together with eliciting itch probably, because of the damage to the skin, elicits considerable nociception (pain) i.e. are the Nptx2-neurons nociceptor activated by pain or are they pruriceptors? The CFA control experiments are not convincing as they 1) the CFA was injected into the paw not into the hairy skin like in the itch studies, 2) the type of pain produced by CFA is very different from that caused by long-term skin damage, and 3) 14 days after CFA, the behavioral sensitization caused by CFA has returned to baseline this is not comparable to the lesions in the skin in the itch models.
- 7) The Nptx2 knockout mice exhibited reduced GRP-neuron activity and itch behavior, but since

Nptx2 is expressed widely in neural tissues this could be caused by loss of Nptx2 in supraspinal circuits which would negate (the studies main proposal of this study) that it serves to increase activity of pruriceptor afferents on GRPR-neurons. To attempt to control for this major concern the study uses an AAV mediated strategy to deliver Nptx2 to DRG neurons in a Nptx2 null background to examine whether this is sufficient to recover the itch sensitization phenotype. This experiment is itself problematic since in order to deliver the AAV, the back skin was incised, and an involved surgery performed to expose the cervical spinal cord (removing muscle). Even at 3 weeks after surgery, this would be expected to cause sensitivity to the neck area and this is in fact shown in the reported results (much lower itch responses). Since the DCP agent were applied to the nape this confounds the study. The number of scratching bouts elicited by control virus in surgical operated Nptx2 null mice is approx. 150 scratch bout/day compared to 500 for non-surgery mice. The difference between control and Nptx2 recovery virus injected mice, by comparison, was 120 versus 150. Added to this issue is the problem that Nptx2 is over-expression in all DRG (versus a small number of neurons in normal mice, figure 3). Therefore, the design of this experiment is extremely poor with multiple possible explanations for the presented data. This experiment needs to be replaced with one using conditional Nptx2 knockout animals for the conclusions proposed in the paper to be believable.

Minor problems

- 1) The traces in figure 1g do not look representative. The summated data in panel h shows a 2x increase in frequency but the traces looks like much more than this. Also, the amplitude of the traces looks like there should be a difference. This applies to the sample traces in figures 2 and 4.
- 2) Numbers of Nptx2 co-localized neurons need to be reported.
- 3) The primary first description of Nptx2 was O'Brien 1999 Neuron should be cited.

re: comments of Reviewer 1

Glutamatergic mechanisms regulating chronic itch are poorly understood. In this manuscript, Kanehisa et. al. demonstrate an essential role of NPTX2 in murine sensory neurons for facilitating excitatory synaptic inputs onto spinal GRPR neurons in two chronic itch models. Their electrophysiological recordings of AAV-GrprP-mCherry-infected spinal dorsal horn neurons indicate that AMPAR-mediated sEPSCs in GRPR neurons were significantly facilitated in mice with chronic itch. Interestingly, NPTX2, an AMPAR-interacting partner, is significantly up-regulated in DRG neurons under chronic itch conditions. Next, they show that NPTX2 is essential for excitatory synaptic facilitation in GrprP-mCherry neurons using loss of function and gain of function approaches. Overall, these findings are an interesting and important. However, major revision is necessary to improve the manuscript.

Reply: We are pleased that Reviewer 1 finds that our findings are “an interesting and important”, and we thank him/her for their detailed suggestions and comments. In the revised manuscript, we have conducted many additional experiments and included the results that address the points raised by this reviewer. We have incorporated the results of these new experiments into the figures and text of the revised manuscript, as indicated below, and we have made all the other changes suggested by the reviewer.

Major concerns

1. The authors should provide the counting data and the percentage of overlaps between NPTX2 and other markers (e.g. CGRP, IB4, TRPV1, NF200). Is GRP expressed in NPTX2 neurons? If so, do GRP/glutamate co-released in these chronic itch models. What is the role of GRP in glutamatergic facilitation of GrprP-mCherry neurons? This should be examined and compared between the control and mice in these pathological conditions.

Reply: As suggested by the reviewer, we have now performed additional immunohistochemical experiments to quantitatively analyze the percentage of overlaps between NPTX2 and other markers of DRG neurons (8 sections from 4 mice). The percentage of neurons positive for the markers CGRP, TRPV1, IB4, and NF200 per total NPTX2⁺ DRG neurons tested were 79.8 ± 5.0% (of 454 neurons tested), 33.3 ± 6.1% (of 454 neurons tested), 9.6 ± 2.7% (of 430 neurons tested), and 35.9 ± 2.5% (of 583 neurons tested), respectively (**Figure 2f**). We have now included these data in the revised figures and the corresponding text (**lines 192–194**).

We conducted a new experiment using RNAscope *in situ* hybridization to examine GRP expression in NPTX2 neurons in the DRG, as there is no available antibody specific for GRP. Consistent with previous reports (e.g., Bell et al., *Sci Rep* 10, 13176, 2020), *Grp* mRNA signals were clearly detected in the SDH, confirming the ability of the RNA probes to detect *Grp* mRNA (**Figure A**, below). In the DRG, however, these signals were not observed in the neurons, including NPTX2⁺ neurons (**Figure A**). Such a low level of *Grp* mRNA expression in the DRG is also consistent with previous data (e.g., Mishra et al., *Science* 340, 968–971, 2013). Thus, our data indicate that *Grp* mRNA is not co-expressed with NPTX2 in DRG neurons.

Figure A. *Grp* mRNA expression in the DRG and SDH.

RNAscope *in situ* hybridization analysis for *Grp* and *Nptx2* mRNA expression in the DRG and SDH of DCP-treated mice. Scale bar, 100 μ m.

2. Although AMPAR antagonist abolished the sEPSCs in *GrprP-mCherry* neurons in DCP-treated mice, and NPTX2 is known as AMPAR-interacting partner, it only implies the possible connection between NPTX2 fibers and *GrprP-mCherry* neurons. The role of glutamate in itch is unclear. Peripheral *Vglut2*-mediated glutamatergic transmission is not required for acute itch (Liu, et al; *Neuron*, 2010). Is *Vglut2* expressed in NPTX2 neurons? Others showed the involvement of glutamate within the dorsal horn microcircuits (Wan et al, 2017, Aresh, et al, 2017). Thus, some evidence supporting the direct synaptic connection between NARP fibers and *GrprP-mCherry* neurons would be necessary to show whether the effect is direct or indirect.

Reply: We performed immunohistochemical experiments to examine VGLUT2 expression in NPTX2⁺ DRG neurons of DCP mice and found co-localization of VGLUT2 and NPTX2 (Figure 2e). Quantitatively, $82.3 \pm 7.1\%$ of total NPTX2⁺ DRG neurons were positive for VGLUT2 immunofluorescence (968 total NPTX2⁺ neurons tested, 9 sections from 3 mice) (Figure 2f) (lines 190–191). Furthermore, to provide more direct evidence for the synaptic connection between NPTX2⁺ terminals and GRPR⁺ neurons in the SDH, we performed an electron microscopy experiment. At the presynapse connected to a postsynaptic GRPR⁺ (*mCherry*⁺) neuron in the SDH (visualized by large immunogold particles conjugating antibody for *mCherry*), we observed small immunogold particles conjugating antibody for NPTX2 (Figure 2h). These new results suggest that NPTX2⁺ terminals form synapses directly with GRPR⁺ SDH neurons and contribute to AMPAR-mediated glutamate synaptic transmission, which strengthens the original conclusion of this manuscript. We have included these new data in the figures and the corresponding text (Abstract, lines 36–37; Results, lines 207–211; Discussion, lines 311–312).

3. Fig. 1d: what is the mean value of the resting potential? Did the authors find a correlation between the resting potential and the firing pattern of *GrprP-mCherry* neurons? The delayed pattern is more common between -70 and -80 mV. Several studies have characterized the electric properties of GRPR neurons, those studies should be cited and discussed. Fig. 1e: was the ability of GRP to induce action potentials dependent on neuron resting potential?

Reply: The mean values of the resting membrane potential (RMP) for each firing pattern in the initial Figure 1d (now Figure 1f) were -65.3 ± 1.3 mV (delayed), -63.7 ± 3.3 mV (transient), and -64.4 ± 2.3 mV (tonic). There were no significant differences

between the groups ($P=0.942$, tonic vs. delay; $P=0.985$, tonic vs. transient; $P=0.908$, delay vs. transient). As pointed out by the reviewer, Pagani et al. have reported that RMP was between -70 and -80 mV (*Neuron* 103, 102–17, 2019). The exact reason for this difference remains unclear, but it may be related to several methodological differences. These include the age of the mice, preparation of spinal cord slices, and/or spinal segments. While Pagani et al. used transverse slices from the lumbar spinal cord segments of young mice (*Neuron* 103, 102–117, 2019), our study used sagittal slices from the cervical spinal cord segments of adult mice. Supporting this notion, recent studies using slices from the cervical spinal cord segment of adult mice reported that the RMP of GRPR⁺ neurons is approximately -62 mV (Liu et al., *PNAS* 116, 27011–27017, 2019; Koga et al., *J Allergy Clin Immunol* 145, 183–191, 2020), which is similar to our data. We have briefly discussed this point with citing these papers in the revised manuscript, as suggested (**lines 95–103**).

In Figure 1e (now Figure 1g), the mean values of the RMP of GRPR⁺ neurons with and without action potentials were -64.4 ± 1.8 mV ($n=5$) and -63.5 ± 1.8 mV ($n=16$), respectively. There was no significant difference between them, indicating that the ability of GRP to induce action potentials is not dependent on the basal RMP of the tested neurons. We have included these data in the Result section (**lines 106–109**).

4. It is quite surprising that the facilitation of both spontaneous and miniature EPSCs consists only in an increase of frequency, with no effects on amplitude. If an increase of clustering of postsynaptic AMPA receptors is involved as the authors implied, I would expect also an increase of amplitude, unless only new AMPA-mediated synapses are involved and no changes in the number of AMPA receptors at the existing synapses occur. Are the cumulative distributions of miniature EPSC amplitudes similar in control and in mice with chronic itch.

Reply: As pointed out, the amplitude also appeared to be increased in DCP-treated mice. According to the minor comment #1 below, we have now performed all the electrophysiological experiments again to obtain the data for a longer duration (10 min). By analyzing the newly obtained data (acetone control group, $n=11$ neurons; DCP group, $n=14$ neurons), we found a significant increase in both the frequency and amplitude of spontaneous EPSCs (**Figure 1i, j**). We also found significant increases in both the frequency and amplitude of miniature EPSCs in acetone- and DCP-treated mice (**Supplementary Figure 1**). Similarly, both the frequency and amplitude of spontaneous EPSCs were also increased in GRPR⁺ SDH neurons of CV-NC/Nga mice compared with SPF-NC/Nga mice (**Figure 1l, m**). In addition, the trimming of the toenails reduced this increased frequency and amplitude of spontaneous EPSCs, although the statistical value for the difference in the amplitude did not reach a significant level ($P=0.053$) (**Supplementary Figure 2c, d**). Importantly, NPTX2 KO suppressed both the frequency and amplitude of spontaneous EPSCs (**Figure 3a, b**). Together, these results indicate that both the frequency and amplitude of glutamatergic synaptic inputs onto GRPR⁺ SDH neurons are facilitated under chronic itch conditions, which are dependent on NPTX2. We have included these new data in the revised manuscript.

5. Fig. 3 shows the critical evidence that chronic itch enhanced the expression of

NPTX2 mRNA and protein in cervical DRG neurons. There are many weakly stained neurons in the control DRG. Considering the absence of staining in NPTX2 KO DRG (Sup. Fig. 2a), the weak staining in control DRG is probably real. Does DCP treatment increase the number of positive neurons, the signal intensities, or both? As mentioned in #1, all the immunostaining results should be quantified and compared.

Reply: Like the reviewer, we considered the possibility that upregulation of NPTX2 might be due to both the increase in the number of NPTX2⁺ DRG neurons and the increase in the signal intensities of NPTX2 immunoreactivity. We counted the number of NPTX2⁺ neurons in the DRG of control and DCP-treated mice and measured the immunofluorescence intensity of NPTX2 on a cell-by-cell basis. Under our criterion that DRG neurons in which the S/N ratio of NPTX2 immunofluorescence was ≥ 4.0 were counted as being positive for NPTX2, the percentage of NPTX2⁺ neurons per total neurons tested was on average 2.9-fold higher in DCP-treated mice than in control mice (**Fig. 2d**). Furthermore, we also found that the mean level of intensity of NPTX2 immunofluorescence per DRG neuron was also 1.4-fold higher in DCP mice. These new data indicate that DCP treatment increases both the number of NPTX2⁺ DRG neurons and the intensity of NPTX2 immunofluorescence in individual DRG neurons. These have been incorporated into the revised figures and corresponding text (Results, **lines 181–183**; Methods, **lines 433–438**).

Minor points

1. The durations of EPSC recording are very short (3 min for sEPSCs and 2 min for mEPSCs). These recordings usually show some variability and a longer duration (at least 10 min) is preferable.

Reply: As described in detail in our response to the major comment #4 above, we have now re-examined all electrophysiological experiments and have obtained spontaneous and miniature EPSC data with a longer duration (10 min). For the data, please see the response described above.

2. Toenail trimming is interesting. How many times was the experiment repeated? What's the n number? The change is minimal comparing to the >2.5 fold increase of NPTX2 mRNA after DCP painting. It's not even close to be prevented by toenail trimming as the authors stated in the discussion.

Reply: In the behavioral and PCR (for *Nptx2* mRNA) experiments, experiments were performed triplicate, and the total number of mice tested was 17 (control, n=8; trimming, n=9) and 17 (control, n=8; trimming, n=9), respectively (the number for PCR experiments has been added in **line 177**). In patch-clamp recordings, we used spinal cord slices from 10 mice (control, n=4; trimming, n=6) and recorded them from 41 neurons (control, n=18; trimming, n=23) in the initial version. We then used spinal cord slices from 6 mice (control, n=3; trimming, n=3) and recorded them from 28 neurons (control, n=12; trimming, n=16) in the revised version. As stated by the reviewer, the effect of toenail trimming on NPTX2 upregulation does not seem to match that on scratching behavior. The difference might be because of the fact that the trimming of the toenails produces effects not only on NPTX2, but also on other NPTX2-independent alterations related to chronic itch. One candidate for the latter is the activation of astrocytes in the SDH, glial cells that are crucial for chronic itch and

are suppressed by toenail trimming (Shiratori-Hayashi et al., *Nat Med* 21, 927–931, 2015). As shown in the initial version, astrocyte activation in SDH was not affected by NPTX2-knockout mice. In revising the manuscript, we have discussed the possibility described above in the Discussion (**lines 276–281**).

3. The scratching numbers vary greatly in different experiments. Control mice in Fig. 5 h and i scratched only about 150 times, which are much fewer than that in Fig. 5a and 2b. Is it caused by different painting procedures?

Reply: In the experiments related to Figure 5h and i in the initial version (now Figure 4h and i), we typically applied DCP only to the left side of the back and counted the number of scratching behaviors by the left hind limb because we injected AAV vectors into only the left spinal nerve, in order to minimize the time and tissue damage related to the operation. Therefore, the number of scratching behaviors in these experiments was approximately half of that in WT mice (Figure 4a) in which DCP was applied to both sides of the back and scratching behavior was counted in both hind limbs. However, we did not describe this in the Method section, which was a mistake. We apologize for this and have now included this information in the Result (**lines 252–255**) and Method sections (**lines 401–402, 486–492**), and the legend of Figure 4.

4. Supplementary Fig. 2e shows immunostaining images in the spinal cord of CV-NC/Nga mice. Were the mice painted with DCP as described in the figure legend?

Reply: The mice were not painted with DCP. We apologize for this error. We have corrected this description in the figure legend.

5. The authors show that the two different chronic itch models are consistent in molecular changes. However, the two models are not identical. Have the authors examined the expression levels of NPTX2 in the lumbar DRGs of CV-NC/Nga mice?

Reply: As requested by the reviewer, our additional immunohistochemical experiments showed that similar to the DCP model, NPTX2 immunofluorescence in the 4th lumbar DRG of CV-NC/Nga mice was also very weak and did not change compared with that of SPF-CV-NC/Nga mice (**Supplementary Figure 3f, and line 205**).

6. What is the rationale for the dose of GRP used ?

Reply: The concentration of GRP (200 nM) used in this study was in accordance with a previous study (Shumyatsky et al., *Cell* 111, 905–918, 2002) and is a submaximal concentration to activate GRPRs in SDH neurons (Hellmich et al., *PNAS* 94: 751–756, 1997; Koga et al., *J Allergy Clin Immunol* 145, 183–191, 2020). We have also confirmed a similar depolarization of GRP at 300 nM (Post – Pre; 8.55 ± 1.25 mV, n=7 neurons), a concentration used in our previous study (Koga et al., *J Allergy Clin Immunol* 145, 183–191, 2020) (**lines 104, and 109–113**).

7. Scale bars should be double checked.

Reply: We have now checked all scale bars in our manuscript.

re: comments of Reviewer 2

This is an interesting report on the role of nptx2 facilitating synaptic transmission between primary afferents and GRP+ spinal dorsal horn neurons in chronic itch conditions. The authors convincingly show that nptx2 is upregulated in primary afferents in two models of chronic itch and that spinal dorsal horn GRP+ neurons investigated in a slice preparation are rendered hyperexcitable under these conditions. Nptx2 was found upregulated in peptidergic primary afferents in the two models and genetic ablation of Nptx2 abolished hyperexcitability and reduced scratching behavior, which was rescued by re-expression of Nptx2 in primary afferents. Also, toenail trimming that reduced scratching-related skin damage including local inflammation and scratching behavior normalized neuronal excitability of spinal GRP+ neurons. The authors suggest that Nptx2 activity dependently released from primary afferents innervating the inflamed skin facilitates spinal processing of itch.

Reply: We are pleased that Reviewer 2 finds that our study is “interesting” and “an important contribution in the field”. As described below, we have performed new experiments to address the points raised by Reviewer 2. We have incorporated the results of these new experiments into the figures and text of the revised manuscript, as indicated below.

When considering the electrophysiological data one would love to see the characteristics of GRP neurons in the itch models (distribution of tonic, transient and delayed firing; membrane potential) – assuming that the chronic itch condition with ongoing peripheral input and GRP release modified the characteristics. If this is the case, the authors need to discuss how this might affect their findings of reduced sEPSP and mEPSP.

Reply: As requested by the reviewer, we have performed patch-clamp recordings to characterize the firing pattern of GRPR⁺ neurons in DCP-treated mice (we think the reviewer meant to ask us to examine GRPR (not GRP) neurons because our study did not focus on GRP neurons). The percentage of GRPR⁺ neurons with each firing pattern (n=27 neurons tested) was 70.4% (delayed: 19/27), 11.1% (transient: 3/27), and 18.5% (tonic: 5/27). The resting membrane potential in DCP-treated mice was slightly higher (-63.2 ± 1.7 mV, n=11 neurons) than that in acetone-treated control mice (control; -67.6 ± 2.0 mV, n=16 neurons), but the difference was not statistically significant (P=0.134, Mann Whitney test). Considering the patch-clamp recordings in which the spinal cord slices were prepared without attachment of DRG soma and their distal and proximal nerves, it is unclear whether GRPR⁺ SDH neurons receive peripherally derived ongoing inputs and GRP signals under our experimental conditions. These data have been included in the text (**lines 125–131**).

Upregulation of Nptx2 in primary afferents was quite broad suggesting that it might not have an effect restricted to pruriceptors, but also involves nociceptors. The authors have included a chronic inflammatory model – however, they tested 14 days after CFA – this appears somewhat late considering the time course of hyperalgesia with behavioral changes being maximum within a few days.

Reply: As suggested by the reviewer, we have performed an additional experiment to

examine the level of *Nptx2* mRNA in the DRG at an early time point (day 3) after CFA injection. *Nptx2* mRNA was not significantly changed (PBS-treated control mice, n=4; CFA-treated mice, n=4) (**Figure B** below). According to the major comment #6 from Reviewer 3, we have deleted our data related to the CFA experiments in the revised manuscript.

Figure B. *Nptx2* mRNA expression in the DRG 3 days after CFA injection.

Nptx2 mRNA in the lumbar DRG in mice 3 days after intraplantar injection of CFA (0.025 mg/5 μ L, n=4) or PBS (n=4). Values represent mean \pm S.E.M.

*Moreover, there are reports on a potential role of *Nptx2* in chronic pain*

<https://doi.org/10.1101/2020.03.19.996645>;

<https://doi.org/10.1016/j.cophys.2019.10.002>

that should be discussed. The comparison might reveal that rather than the difference between itch and pain there may be a crucial difference between acute and chronic sensitization. This would shed some light on the general problem we are facing when acutely evoked pain or itch models are used to investigate chronic conditions that are dominated by spontaneous itch/pain.

Reply: Although the reports that the reviewer has listed above are unpublished preliminary findings and are not peer-reviewed, our data clearly demonstrated that the levels of NPTX2 expression are much higher in DRG neurons than in the spinal cord. Furthermore, our new experiments using RNAscope *in situ* hybridization have also revealed only very faint signals of *Nptx2* mRNA in the SDH of DCP-treated mice (**Supplementary Figure 3d**, and **lines 203–204**), which was in stark contrast to those in the DRG. Moreover, as mentioned in the original manuscript, NPTX2 deficiency has been demonstrated to not suppress chronic pain (Miskimon et al., *J Neuroimmunol* 274, 86–95, 2014). However, we cannot completely exclude a putative role of NPTX2 expressed in SDH neurons. Because these data and report cited by the reviewer, again, are only unpublished preliminary findings without peer reviews, we have only added a brief discussion regarding this point in the Discussion section (**lines 291–292**).

re: comments of Reviewer 3

The manuscript by Kanehisa et al. entitled “Neuronal pentraxin 2 is required for facilitating excitatory synaptic inputs onto spinal itch-transmission neurons under chronic itch conditions” describes work where the contribution of the pentraxin 2 protein to itch responses was examined. Pentraxin 1 and 2 have been previously reported to mediate the uptake of synaptic macromolecules and serve functions in synaptic plasticity. In particular, pentraxin 2 (Narp or NP2 or Nptx2) has been described as an activity-regulated protein which is secreted forming clusters with itself that co-aggregate with the AMPA receptor (O’Brien 1999 Neuron). To understand the potential mechanism, in the spinal cord, underlying chronic itch, the activity of GRPR neurons in the spinal cord was examined. As expected, activity was increased in the two models of chronic itch used. Global knockout of Nptx2 attenuated GRPR activity, moderately reduced itch behavioral responses, and improved skin condition in the DCP model of chronic itch. Rescue experiments where Nptx2 was expressed in DRG neurons in Nptx2 null mice are reported to recapitulate the increased itch sensitization to DCP treatment seen in wild-type mice.

The evidence provided for the proposed mechanism for Nptx2 mediate sensitization is insufficient with the tools used in the study inadequately characterized (point 1), unsatisfactory characterization of DRG neurons expressing described (point 5 & 6), and the design of experiment extremely poor and consequently of little use.

Reply: We thank the reviewer for the comments regarding our manuscript and for the detailed suggestions. As described below, we have performed many new experiments to address the points raised by Reviewer 3. These results from these experiments have been incorporated into the figures and text of the revised manuscript, as indicated.

Major Problems

1) The study uses of an undefined DNA element upstream of the GRPR gene to drive expression of fluorescent markers and of a DREADDq chemogenetic receptor in spinal cord neurons. This method for identifying and examining the function of GRPR is not adequately tested. There needs to be experiments performed to directly demonstrate that GRPR-neurons are targeted, for instance, by ISH should be used to examine the expression of transgene versus GRPR. Also, the numbers and type of neurons reported here are at odds with those reported by Pagani et al, (2019 Neuron), there were inhibitory as well as excitatory neurons described in the latter paper.

Reply: Thank you for your suggestion. We have now performed new experiments using RNAscope *in situ* hybridization and found that SDH neurons targeted by the AAV-GrprP vector used in this study express *Grpr* mRNA (**Figure 1d**). Furthermore, GRP-induced depolarization in the SDH neurons targeted by the AAV-GrprP vector were suppressed in GRPR KO mice, strongly supporting our notion that SDH neurons targeted by the AAV-GrprP vector express functional GRPRs. These new data have been incorporated into the revised figure and the corresponding text (**lines 87–88, 109–113**).

The reviewer noted that the numbers and types of neurons in our manuscript are at odds with those reported by Pagani et al. (*Neuron* 3, 102–117, 2019), but the percentage of GRPR⁺ neurons in the SDH in their study was 58% and 21% of the recorded neurons

and showed delayed and tonic firing patterns, respectively, which are similar to those of GRPR⁺ neurons in our study (delayed type, 63.6%; tonic type, 27.3%). While Pagani et al. reported that 27% of GRPR⁺ neurons tested were positive for PAX2, our additional immunostaining experiments and quantitative analysis revealed that 18.3% of GRPR⁺ neurons tested were positive for PAX2 (176 neurons per 902 total GRPR⁺ neurons tested; 12 sections from 4 mice). We have included this data and description in the revised manuscript (**lines 88–90**).

2) CNQX is a potent AMPA/kainite receptor antagonist but is also an antagonist at the glycine modulatory site on the NMDA receptor complex. A comment needs to be added to address this as glycine release from inhibitory IN might be also affected with CNQX.

Reply: We tested the effect of NBQX, a specific antagonist of AMPAR, and found that this antagonist abolished the EPSCs (**Figure 1k**, and **line 139**), which supports our conclusion that the observed EPSCs are mediated by AMPARs.

3) There is considerable controversy about whether primary afferents synapse directly on GRPR-neurons. The paper assumes the latter but provides not direct evidence for this interaction. At a minimum this point should be addressed in a fair way in the Discussion.

Reply: Recent electrophysiological data have shown that GRPR⁺ SDH neurons receive excitatory monosynaptic inputs from primary afferent C fibers (Bardoni et al., *Sci Rep* 9, 15804, 2019). We have cited this paper in the revised manuscript (**line 304**). Furthermore, as mentioned in our response to the major comment #2 from Reviewer 1, our new experiment using electron microscopy revealed that NPTX2 was observed at presynaptic terminals on GRPR⁺ neurons in the SDH (**Figure 2h**). Considering our data, which showed that NPTX2 is highly expressed in DRG neurons but not in SDH neurons (**Figure 2g and Supplementary Figure 3d**), it is reasonable to suggest that NPTX2⁺ primary afferents form synapses directly with GRPR⁺ SDH neurons. Thus, we have incorporated these new data in the figure and the text (Abstract, **lines 36–37**; Results, **lines 207–211**; Discussion, **lines 311–312**).

4) Figure 2 presents data that could easily be combined with the data in figure 1 or move to the supplements.

Reply: We have moved the data in the initial Figure 2 to **Supplementary Figure 2**, as suggested.

5) The level of Nptx2 is shown to be increased in DRG neurons in Figure 3. Nptx2 is expressed in many subtypes of DRG neurons, see Sharma et al., 2020 Nature [https://kleintools.hms.harvard.edu/tools/springViewer_1_6_dev.html?datasets/Sharma2019/alland Usoskin et al, Nature Neurosci 2015](https://kleintools.hms.harvard.edu/tools/springViewer_1_6_dev.html?datasets/Sharma2019/alland%20Usoskin%20et%20al,%20Nature%20Neurosci%202015); <http://linnarssonlab.org/drg/> Given these published results, why, under normal conditions, is there no detectable Nptx2 expression in DRG under naïve conditions?

Reply: First, we would like to mention that the levels of NPTX2 immunoreactivity were low, but were still detectable in our experiments as shown in Figure 3c (now Figure 2c). This weak immunoreactivity is not non-specific because there is no detectable NPTX2 immunofluorescence in NPTX2 KO mice without DCP treatment (**Supplementary**

Figure 3a). The intensity of NPTX2 immunofluorescence in our study was relatively weak compared with that in the previous paper (Miskimon et al. *J Neuroimmunol* 274, 86–95, 2014), but this may be due to experimental conditions; we acquired immunofluorescence images using a confocal laser microscope, in which, for quantitative analysis, the intensity levels of NPTX2 immunofluorescence of acquired images were not saturated, even if the level increased in the chronic itch model. In revising the manuscript, we have included this in the Method section (**lines 432–433**).

6) In addition (Figure 3), the colocalization studies were performed with markers that have little relevance to itch. Recent work demonstrate that the NP2 and NP3 populations of neurons are responsible for itch. CGRP, TrkA, Trpv1, and IB4 are not selective for these populations of neurons. This cast serious doubts about whether Nptx2 is overexpressed in the appropriate classes of neurons. Indeed, a result of the reported chronic itch are skin lesions which together with eliciting itch probably, because of the damage to the skin, elicits considerable nociception (pain) i.e. are the Nptx2-neurons nociceptor activated by pain or are they pruriceptors? The CFA control experiments are not convincing as they 1) the CFA was injected into the paw not into the hairy skin like in the itch studies, 2) the type of pain produced by CFA is very different from that caused by long-term skin damage, and 3) 14 days after CFA, the behavioral sensitization caused by CFA has returned to baseline this is not comparable to the lesions in the skin in the itch models.

Reply: As stated by the reviewer, we also considered that the upregulation of NPTX2 expression does not occur in specific subpopulations of DRG neurons. However, as shown in the initial version, NPTX2 immunofluorescence was selectively observed in CGRP⁺ or TRKA⁺ DRG neurons under chronic itch conditions. Our additional quantitative analysis revealed that approximately 80% of NPTX2⁺ DRG neurons were CGRP-positive (**Figure 2f**). Thus, our data indicate that NPTX2 upregulation occurs selectively in CGRP⁺ DRG neurons. A previous study has demonstrated that the NP2 population expresses CGRP and TRKA (Usoskin et al., *Nat Neurosci* 18, 145–153, 2019) and that CGRP⁺ primary afferents elongate into the epidermis of the itchy skin (Andoh et al., *Neurosci Lett* 672, 84–89, 2018) and contribute to itch transmission (McCoy et al., *Neuron* 78, 138–151, 2013), supporting our notion that NPTX2 expressed in these DRG neurons plays a pivotal role in chronic itch. Consistently, our behavioral data showed suppression of chronic itch either by knocking out NPTX2, or by ectopically expressing dnNPTX2 in DRG neurons, and rescue of the itch-related behavioral phenotype in NPTX2 KO mice by DRG neuron-specific ectopic expression of wild-type NPTX2. Although CGRP has been shown to be expressed in other populations of DRG neurons (e.g., PEP1 and 2), Miskimon et al. have clearly demonstrated that NPTX2 KO mice exhibit no influence on behavioral responses in acute and chronic pain models (*J Neuroimmunol* 274, 86–95, 2014). Furthermore, if NPTX2 plays a role in facilitating nociceptive pain transmission, its deficiency enhances itch-related behaviors, as nociceptive signals are known to inhibit pruriceptive transmission in the SDH. However, our data showed that NPTX2 KO suppressed itch-related behavior. Moreover, to distinctly assess itch- and pain-related behaviors in DCP-treated mice, we utilized a cheek model in which DCP was topically applied to the cheek and found that the DCP treatment allowed mice to scratch (itch), but not wipe

(pain) their cheek (**Figure C**, below). These new data support our conclusion that NPTX2 plays a selective role in itch transmission under chronic itch conditions. Although the role of NPTX2 in a specific population of DRG neurons in chronic itch remains unclear, this will require a complete study on its own, because it requires new experimental tools enabling conditional knockout of NPTX2 in a population-specific manner, which is not currently available. In the revised manuscript, we have briefly mentioned this point in the Discussion section as an important subject in the future (**lines 287–302**).

In experiments using the inflammatory pain model by CFA, we confirmed that mechanical pain hypersensitivity was observed on day 14 after CFA injection. In addition, we have also shown that NPTX2 expression in DRG neurons does not change 3 days after CFA injection, a time point when behavioral pain hypersensitivity develops (**Figure B**, please see our response to the second comment from Reviewer 2). However, as stated by the reviewer, we agree that the type of pain produced by CFA is very different from that caused by long-term skin damage. Thus, we have deleted our data related to the CFA experiments in the revised manuscript.

Figure C. Topical application of DCP to the cheek produces scratching, but not wiping behavior.

DCP or vehicle (acetone) was topically applied to the shaved cheek, and, 14 days later, scratching behavior by hind limbs and wiping behavior by fore limbs directed toward the DCP-treated cheek were video-taped for 30 min and counted. Vehicle, n=11; DCP, n=11. *** $P < 0.001$. Values represent mean \pm S.E.M.

*7) The *Nptx2* knockout mice exhibited reduced GRP-neuron activity and itch behavior, but since *Nptx2* is expressed widely in neural tissues this could be caused by loss of *Nptx2* in supraspinal circuits which would negate (the studies main proposal of this study) that it serves to increase activity of pruriceptor afferents on GRPR-neurons. To attempt to control for this major concern the study uses an AAV mediated strategy to deliver *Nptx2* to DRG neurons in a *Nptx2* null background to examine whether this is sufficient to recover the itch sensitization phenotype. This experiment is itself problematic since in order to deliver the AAV, the back skin was incised, and an involved surgery performed to expose the cervical spinal cord (removing muscle). Even at 3 weeks after surgery, this would be expected to cause sensitivity to the neck area and this is in fact shown in the reported results (much lower itch responses). Since the DCP agent were applied to the nape this confounds the study.*

*The number of scratching bouts elicited by control virus in surgical operated *Nptx2* null mice is approx. 150 scratch bout/day compared to 500 for non-surgery mice. The difference between control and *Nptx2* recovery virus injected mice, by comparison, was 120 versus 150. Added to this issue is the problem that *Nptx2* is over-expression in all DRG (versus a small number of neurons in normal mice, figure 3). Therefore, the design of this experiment is extremely poor with multiple possible explanations for the*

presented data. This experiment needs to be replaced with one using conditional Nptx2 knockout animals for the conclusions proposed in the paper to be believable.

Reply: One concern raised by this reviewer is that the surgery of the back skin for the microinjection of AAV vectors could have a non-specific effect on scratching behavior. To address this issue, we compared the number of scratching behaviors between groups with and without the injection of control AAV vector (AAV-ESYN-GFP) into the left spinal nerve, and demonstrated that the surgery and AAV injection themselves did not affect scratching behavior (**Supplementary Figure 4**). These data imply that the surgery for AAV injection has no influence on itch-related behavioral responses under our experimental conditions (**lines 260–264**). As noted by the reviewer, the number of scratching behaviors elicited by control AAV vector in NPTX2 KO mice was approximately half of that observed in WT mice without AAV injection. As described in our response to the minor comment #3 from Reviewer 1, we injected AAV vectors into the left side spinal nerve only and therefore applied DCP only on the skin of the left back, and counted only the number of scratching behaviors by the left hind limb. This is the reason why the number of scratching behaviors in the experiments with AAV vector injection is much smaller than that in other experiments. However, this method was not described in the Methods section of the initial version. We apologize for this mistake. We have corrected this in the Result (**lines 252–255**) and Method sections (**lines 401–402, 486–492**), and the legend of Figure 4.

We therefore suggest that the experiments we designed using AAV vectors are an appropriate strategy for investigating the role of NPTX2 in DRG neurons. Thus, based on our data showing that chronic itch is suppressed by expressing dnNPTX2 in DRG neurons and that suppression of scratching behavior in NPTX2 KO mice was rescued by DRG neuron-specific expression of NPTX2, it is reasonable to conclude that NPTX2 expression in these DRG neurons plays a pivotal role in chronic itch. We believe that the reviewer will agree with our explanation based on these new control data, without additional data using genetic tools for conditional knockout (e.g., *Nptx2*^{flox/flox} mice) that have not been previously reported.

Minor problems

1) The traces in figure 1g do not look representative. The summated data in panel h shows a 2x increase in frequency but the traces looks like much more than this. Also, the amplitude of the traces looks like there should be a difference. This applies to the sample traces in figures 2 and 4.

Reply: According to minor comment #1 from Reviewer 1, we have newly obtained all electrophysiological data with recording for a longer duration (10 min). As a result, both the frequency and amplitude of spontaneous EPSCs were significantly increased in chronic itch models (**Figures 1i, 1j, 1l, 1m, 3a, 3b, and Supplementary Figures 1 and 2c, d**). We have replaced the initial data with new data in the revised manuscript.

2) Numbers of Nptx2 co-localized neurons need to be reported.

Reply: We have quantitatively analyzed the co-localization of NPTX2 and neuronal markers (**Figure 2f**), as described in our response to the major comment #1 from Reviewer 1.

3) *The primary first description of Nptx2 was O'Brien 1999 Neuron should be cited.*

Reply: We have included this paper in the revised manuscript (**line 74**).

REVIEWER COMMENTS

Reviewer #1 (Remarks to the Author):

In this revised study, Kanehisa et al. have provided additional evidence that supports the role of glutamatergic transmission in chronic itch and the connections between Nptx2 fibers and GRPR neurons. Electrophysiological evidence is solid and convincing. However, one major issue remains to be resolved. A recent study has demonstrated that spinal Grp neurons are dispensable for itch behaviors and GRP from DRGs is a crucial neuropeptide for itch (Barry et al, 2020). The lack of Grp mRNA signal in DRGs using RNAscope could be caused by poor sensitivity, since the image provided in the reply letter does not look like typical Grp ISH in the spinal cord, which should be very robust and highly concentrated in lamina II. The authors overlooked many studies demonstrating positive results regarding Grp mRNA in DRGs, raising concerns about their conclusion.

It is also untrue to say that “no specific antibody for GRP”. The specificity of the GRP antibody has been demonstrated with several methods by multiple labs (Takanami et al. 2014, Fleming et al. 2012, Barry et al. 2016, Solorzano et al (2015). Solorzano et al. also demonstrated that the antibody is specific. Otherwise, they would not be able to claim that the source of GRP lies in the spinal cord.

Reduced itch is rather modest in the DCP model and a GRPR antagonist can be used to determine whether the remaining behavior is mediated by GRP. Immunostaining of GRP in DRGs and spinal cord under normal, chronic itch condition and Nptx2 KO mice should be performed. Other approaches such as qRT-PCRs, western blots and dorsal rhizotomy can also be used to examine this issue. This will clarify the contribution of GRP vs. glutamate to activation of GRPR neurons.

The authors indicated that spinal Grp neurons are also innervated by Nptx2 afferents and release GRP/glutamate (Suppl. Fig. 6). However, spinal Grp neurons are potently activated by painful stimuli (Sun et al, 2017), which is supported by a recent study showing that MrgprA3 neurons relay nociceptive information to the spinal cord (Sharif et al, 2020). The model proposing that a direct Nptx2-GRPR neural pathway to activate GRPR neurons first via glutamate and an indirect pathway through Grp neurons to release GRP/glutamate do not fit with what we know about a subset of spinal Grp neurons.

Vglut2 and Nptx2 are largely co-expressed, but others found that glutamate in Vglut2 neurons is dispensable for itch and deletion of Vglut2 even enhanced itch (Liu et al. 2010; Legerstrom et al 2010), a conclusion obviously at odds with the present study. A recent peer-reviewed paper found that Nptx2 KO mice exhibited profound nociceptive allodynia in a neuropathic pain model (Wang et al. 2021). This suggests that glutamate from Nptx2 fibers may also play a role in nociceptive transmission, possibly through spinal Grp neurons (?).

These discrepancies warrant a careful analysis of itch and pain behavior of Nptx2 KO mice. Acute itch (histamine and chloroquine) of Nptx2 KO mice should be examined, which would help to determine to what degree that glutamate from Nptx2 neurons contributes to itch (as compared to Vglut2 CKO and Grp CKO mice). Moreover, a comparison of pain behaviors with that of Vglut2 CKO mice may be important to address these conflicting results.

Overall, the interpretation of the results and the claim that the spinal cord GRP as a secondary pathway for itch are inconsistent with recent studies. The relationship between GRP, glutamate and GRPR neurons need to be thoroughly examined and discussed.

Others:

1. Are there other Nptxs in DRGs may compensate for the loss of Nptx 2 in DRGs?
2. Is DCP itch histamine-dependent ?

Reviewer #3 (Remarks to the Author):

The revised manuscript is much improved. I commend the authors on their willingness to perform the additional studies to confirm their interpretations.

re: comments of Reviewer 1

In this revised study, Kanehisa et al. have provided additional evidence that supports the role of glutamatergic transmission in chronic itch and the connections between Nptx2 fibers and GRPR neurons. Electrophysiological evidence is solid and convincing. However, one major issue remains to be resolved. A recent study has demonstrated that spinal Grp neurons are dispensable for itch behaviors and GRP from DRGs is a crucial neuropeptide for itch (Barry et al, 2020). The lack of Grp mRNA signal in DRGs using RNAscope could be caused by poor sensitivity, since the image provided in the reply letter does not look like typical Grp ISH in the spinal cord, which should be very robust and highly concentrated in lamina II. The authors overlooked many studies demonstrating positive results regarding Grp mRNA in DRGs, raising concerns about their conclusion.

Reply: We are pleased that Reviewer 1 finds that our additional electrophysiological data are ‘solid and convincing’, and we thank him/her for their further suggestions and comments.

One concern raised by the reviewer is associated with our data on RNAscope *in situ* hybridization. To improve the detection of fluorescence signals and to determine the laminar localization in the SDH, we have now additionally performed RNAscope experiments using probes for *Grp* and *Prkcg* (encoding protein kinase C γ that is selectively expressed in lamina II neurons). We observed clear and easily recognized signals of *Grp* mRNA that were predominantly located in lamina II (**Figure Aa**, below). This is consistent with previous data reported in several papers (e.g., *Science* 340, 968–971, 2013; *Neuron* 78, 312–324, 2013; *J Neurosci* 35, 648–657, 2015), confirming the ability of the RNA probe to detect *Grp* mRNA. Under the experimental conditions, however, the fluorescence signals of *Grp* mRNA, as observed in the SDH, were not found in the DRG of either vehicle (acetone) or DCP-treated mice (**Figure Ab**). This difference in the expression between the two regions seems to be supported by our data using quantitative PCR experiments (**Figure Ac**) and previous data measured by RNAseq, qPCR, or *in situ* hybridization (e.g., *Science* 340, 968–971, 2013; *J Neurosci* 35, 648–657, 2015; *Nat Neurosci* 18, 145–153, 2015). Consistent with the data in the previous version of our manuscript, this additional experiment also confirmed that *Nptx2* mRNA was detected in DRG neurons, and its level was higher in DCP-treated mice (**Figure Ab**). Therefore, it appears that even if *Grp* mRNA is expressed in DRG neurons including NPTX2⁺ neurons, its levels might be very low (compared with that in SDH neurons).

Figure A. *Grp* mRNA expression in the DRG and SDH.

a,b RNAscope *in situ* hybridization analysis for *Grp* and *Prkcg* mRNA in the SDH (**a**), and *Grp* and *Nptx2* mRNA in the DRG (**b**) of vehicle (acetone)- and DCP-treated mice. Scale

bar, 100 μ m. **c**, Quantitative RT-PCR analysis of *Grp* mRNA (relative to the value of SDH) in the SDH and DRG of WT mice (n=9 mice). Values represent mean \pm S.E.M.

It is also untrue to say that “no specific antibody for GRP”. The specificity of the GRP antibody has been demonstrated with several methods by multiple labs (Takanami et al. 2014, Fleming et al. 2012, Barry et al. 2016, Solorzano et al (2015). Solorzano et al. also demonstrated that the antibody is specific. Otherwise, they would not be able to claim that the source of GRP lies in the spinal cord.

Reply: We apologize for the confusion in this sentence. As correctly noted by the reviewer, Solorzano et al. have shown the specificity of a GRP antibody (commercially available) for the immunohistochemical detection of GRP (*J Neurosci* 35, 648–657, 2015). In their study, immunofluorescence by the GRP antibody was detected in the SDH and DRG of WT mice; however, in GRP KO mice, this disappeared only in the SDH, implying that immunofluorescence by the GRP antibody observed in the DRG seems to be unrelated to GRP expression levels (*J Neurosci* 35, 648–657, 2015). Furthermore, because the host species of this GRP antibody is the rabbit (which is the same as that of NPTX2 antibody used), it is difficult to perform double-immunostaining of NPTX2 and GRP using these two antibodies, both of which are the only antibodies whose specificities for each protein have been validated. Therefore, for the experiments to examine the expression of GRP and NPTX2 in the DRG, we employed RNAscope *in situ* hybridization (**Figure A**).

The pathway from NPTX2⁺ primary afferents to GRPR⁺ neurons via GRP⁺ SDH neurons is depicted in Supplementary Figure 6 as another hypothesis to explain our data. However, although we have demonstrated the synaptic connection of NPTX2⁺ primary afferents to GRPR⁺ neurons, there is no experimental evidence for the direct connection of NPTX2⁺ primary afferents to GRP⁺ neurons in the SDH. Thus, we have deleted the sentence describing that primary afferent-derived NPTX2 may also target GRP⁺ neurons in the Discussion section and the legend of Supplementary Figure 6 (now **Supplementary Figure 7**), and also modified the schematic illustration (**Supplementary Figure 7**).

Reduced itch is rather modest in the DCP model and a GRPR antagonist can be used to determine whether the remaining behavior is mediated by GRP. Immunostaining of GRP in DRGs and spinal cord under normal, chronic itch condition and Nptx2 KO mice should be performed. Other approaches such as qRT-PCRs, western blots and dorsal rhizotomy can also be used to examine this issue. This will clarify the contribution of GRP vs. glutamate to activation of GRPR neurons.

Reply: As pointed out by the reviewer, NPTX2 KO mice did not completely suppress DCP-induced scratching behavior. It is thus possible that others, including GRP signaling, may be involved in the residual scratching behavior in DCP-treated NPTX2 KO mice. One approach for examining this prediction could be to test the effect of a GRPR antagonist, as suggested by the reviewer, but this experiment seems to be difficult in our experiments for the following reasons. Unlike acute itch models, we evaluated chronic itch by counting scratching behavior for 24 hours, and thus, to determine the role of GRP, spinal GRPRs must be blocked during the testing period. At present, commercially available antagonists specific for GRPR, such as RC-3095 (a pseudopeptide antagonist), have been used in other reports (e.g., *J Pharmacol Exp Ther* 337, 822–829, 2011), but these peptide compounds seem to be short-acting (*Regul Pept* 41, 185–193, 1992). Considering this pharmacokinetic property, a number of intrathecal injections are needed for long-term blockade of spinal GRPRs. However, manipulations associated with many intrathecal injections (e.g., holding the mouse) would disrupt mouse behavior, which hampers accurate pharmacological evaluation of

scratching. Another approach to address this comment could be the use of NPTX2 and GRP double KO mice. However, because we do not currently have a GRP KO mouse line, it will take more than one year to generate the double KO mice and to prepare a sufficient number of mice for behavioral tests. We agree that it is important to explore the contribution of GRP vs. glutamate to the activation of GRPR⁺ SDH neurons, but we consider this beyond the scope of our present study, whose initial version focused on the role of NPTX2 in glutamate transmission in the SDH and chronic itch because our additional data have shown that (1) *Grp* mRNA was expressed at a low level in the DRG including NPTX2 neurons (as described in our responses to above comments; **Figure A**) and (2) the levels of *Grp* mRNA in the DRG and SDH were not different between WT and NPTX2 KO mice (**Figure B**, below). Therefore, it seems appropriate to investigate this separately as another study on its own. In revising the manuscript, we have included this issue as an important subject in the future (**lines 350–353, 360–361**).

Figure B. *Grp* mRNA expression in the DRG and SDH of DCP-treated WT and NPTX2 KO mice.

Quantitative RT-PCR analysis of *Grp* mRNA (relative to the value of SDH of WT mice) in the SDH and DRG of DCP-treated WT and NPTX2 KO mice (n=6 mice). Values represent mean \pm S.E.M.

The authors indicated that spinal Grp neurons are also innervated by Nptx2 afferents and release GRP/glutamate (Suppl. Fig. 6). However, spinal Grp neurons are potentially activated by painful stimuli (Sun et al, 2017), which is supported by a recent study showing that MrgprA3 neurons relay nociceptive information to the spinal cord (Sharif et al, 2020). The model proposing that a direct Nptx2-GRPR neural pathway to activate GRPR neurons first via glutamate and an indirect pathway through Grp neurons to release GRP/glutamate do not fit with what we know about a subset of spinal Grp neurons.

Reply: As described in the response above, we have demonstrated that NPTX2⁺ primary afferents directly innervate GRPR⁺ SDH neurons, and there is no direct evidence for the connection between NPTX2⁺ afferents and GRP⁺ SDH neurons. In addition, in line with this, as noted by the reviewer, GRP⁺ SDH interneurons have recently been reported to also be activated by painful stimuli and that the main focus of the present study was the role of NPTX2 in glutamate transmission in GRPR⁺ SDH neurons but not GRP⁺ SDH neurons. Therefore, we have deleted the sentence describing that primary afferent-derived NPTX2 may also target GRP⁺ neurons in the Discussion section and the legend of Supplementary Figure 6 (now **Supplementary Figure 7**), and also modified the illustration in **Supplementary Figure 7**.

Vglut2 and Nptx2 are largely co-expressed, but others found that glutamate in Vglut2 neurons is dispensable for itch and deletion of Vglut2 even enhanced itch (Liu et al. 2010; Legerstrom et al 2010), a conclusion obviously at odds with the present study. A recent peer-reviewed paper found that Nptx2 KO mice exhibited profound nociceptive allodynia in a neuropathic pain model (Wang et al. 2021). This suggests that glutamate from Nptx2 fibers may also play a role in nociceptive transmission, possibly through spinal Grp neurons (?).

Reply: Thank you for your comment. We agree that we should include an explanation

of this important point. Studies by us and others have demonstrated that GRPR⁺ SDH neurons monosynaptically receive AMPAR-mediated glutamatergic inputs from primary afferents (*Mol Pain* 7, 47, 2011; *Sci Rep* 9, 15804, 2019; our present study) and that DCP-induced scratching is suppressed by intrathecal injection of a selective AMPAR antagonist (*Biochem Biophys Res Commun* 533, 1102–1108, 2020). Conversely, as noted by the reviewer, mice lacking VGLUT2 in primary afferent neurons have been reported to exhibit spontaneous scratching behavior (e.g., *Neuron* 68, 529–542, 2010; *Neuron* 68, 543–556, 2010), although this scratching phenotype seems to be inconsistent between VGLUT2 cKO mouse lines used in these studies and is not observed in mice lacking VGLUT2 from nociceptors (*PNAS* 107, 33396–22301, 2010). However, single-cell RNA sequencing has shown that VGLUT2 is broadly expressed in DRG neurons, including the NP2 population that is related to itch (*Nat Neurosci* 18, 145–153, 2015), and, importantly, whether VGLUT2 in WT mice contributes to glutamatergic transmission from primary afferents to GRPR⁺ SDH neurons under normal and chronic itch conditions remains entirely unknown. Furthermore, if NPTX2 enhances VGLUT2-mediated glutamate signaling that can inhibit itch and produce pain, then DCP-treated mice in which NPTX2 is upregulated could display behavioral responses related to pain rather than itch. However, our data showed that mice treated with DCP in their cheek exhibited scratching but not wiping, the latter being an index of pain-like behavior (we have now confirmed NPTX2 upregulation in trigeminal ganglion neurons; **Supplementary Figure 6**). Moreover, NPTX2 KO has been demonstrated to have no effect on pain-like behavior in acute and chronic pain models (*J Neuroimmunol* 274, 86–95, 2014) (although the reviewer mentions that NPTX2 KO mice exhibited profound nociceptive allodynia in a neuropathic pain model, their study (Wang et al. *Neuroreport* 32, 274–283, 2021) did not show such data). Therefore, it is conceivable that under chronic itch conditions, NPTX2 upregulation in DRG neurons, including VGLUT2⁺ neurons, could preferentially strengthen glutamatergic synaptic inputs onto GRPR⁺ itch transmission neurons, although the detailed mechanism needs to be investigated in the future.

Nevertheless, we cannot exclude the possibility that under chronic itch conditions, NPTX2 is also upregulated in a VGLUT2⁺ nociceptor subpopulation that can inhibit itch. If this occurs, why does NPTX2 KO suppress chronic itch rather than enhance it? The underlying mechanism remains unclear, but given that glutamate released from VGLUT2⁺ nociceptors might suppress itch responses by feedforward inhibition via inhibitory neurons in the SDH (*Neurosci Bull* 28, 91–99, 2012; *Nat Neurosci* 17, 175–182, 2014), a plausible explanation could be that the inhibitory SDH neurons cause dysfunction under chronic itch conditions, which results in reduced neuronal excitation by VGLUT2⁺ nociceptor-derived glutamate even if NPTX2 coexists. Indeed, a loss of feedforward inhibition by inhibitory SDH neurons causes an excessive scratching phenotype (e.g., *Neuron* 65, 886–898, 2010; *Nat Neurosci* 21, 707–716, 2018). A failure of nociceptive stimuli (e.g., scratching) to suppress itch has also been reported in patients with atopic dermatitis (*Br J Dermatol* 158, 78–83, 2008).

We have now included the behavioral and immunohistochemical data obtained from mice treated with DCP in their cheek in **Supplementary Figure 6** and the above explanation in the Discussion section (**lines 287–332**).

These discrepancies warrant a careful analysis of itch and pain behavior of Nptx2 KO mice. Acute itch (histamine and chloroquine) of Nptx2 KO mice should be examined, which would help to determine to what degree that glutamate from Nptx2 neurons contributes to itch (as compared to Vglut2 CKO and Grp CKO mice). Moreover, a

comparison of pain behaviors with that of Vglut2 CKO mice may be important to address these conflicting results.

Reply: In the initial version of our manuscript, we have already shown that scratching behavior evoked by intradermal injection of chloroquine was indistinguishable between WT and NPTX2 KO mice (now Figure 4e). In addition, NPTX2 KO had no effect on scratching evoked by intradermal compound 48/80 (Figure 4e), as a model of histamine-dependent acute itch (*Eur J Pharmacol* 351, 1–5, 1998). Thus, NPTX2 is not involved in the behavioral response associated with acute itch. This is consistent with our immunohistochemical and electrophysiological data showing that NPTX2 expression in normal mice is low and that NPTX2 KO does not influence the frequency or amplitude of EPSCs in GRPR⁺ SDH neurons.

A recent study has demonstrated that intrathecal administration of peramppanel, a selective AMPAR antagonist, suppressed the scratching behavior of DCP-treated mice (*Biochem Biophys Res Commun* 533, 1102–1108, 2020), which supports our findings that NPTX2 strengthens AMPAR-mediated glutamate transmission in GRPR⁺ itch transmission neurons. Regarding pain, as described above, NPTX2 KO has already been demonstrated to have no effect on pain behaviors in acute and chronic pain models (*J Neuroimmunol* 274, 86–95, 2014). As described in the Discussion section of the first revision, because the upregulation of NPTX2 expression seems to occur in several subpopulations of DRG neurons under chronic itch conditions, an important subject in the future will be to determine the functional role of glutamate transmission from each subpopulation of NPTX2⁺ DRG neurons in chronic itch and other modalities, including pain. However, a new experimental tool is required to enable conditional knockout of VGLUTs in a NPTX2-expressing DRG neuron population (e.g., *Nptx2-CreERT2* mice), which has not yet been established. Therefore, in revising the manuscript, we have mentioned this as an important future subject in the Discussion section (**lines 287–332**).

Overall, the interpretation of the results and the claim that the spinal cord GRP as a secondary pathway for itch are inconsistent with recent studies. The relationship between GRP, glutamate and GRPR neurons need to be thoroughly examined and discussed.

Reply: As described in our response above, we addressed the issues raised by the reviewer, with new data from additional experiments (**Figures A and B** included in this document, and **Supplementary Figure 6**) and discussion (**lines 287–332, 350–361**). Because of experimental limitations and unavailable tools, the relationship between GRP, glutamate, and GRPR neurons under chronic itch conditions is not completely elucidated at this time, but we consider that this is the next important subject for future research. We believe that the reviewer will agree with our explanation described above, without additional data that will require extensive experiments by developing a new GRP-specific antibody used in the DRG immunohistochemically and by developing or obtaining new mouse lines, generating a sufficient number of mice, and performing behavioral tests.

Others

1. Are there other Nptxs in DRGs may compensate for the loss of Nptx 2 in DRGs?

Reply: We performed additional quantitative RT-PCR experiments to examine the expression of *Nptx1* mRNA in the DRG of WT and NPTX2 KO mice with DCP treatment and found no difference between the two genotypes. These new data have been incorporated in the text (**line 183–185**).

2. Is DCP itch histamine-dependent ?

Reply: Previously, it was reported that mice lacking histamine by disrupting the gene encoding histidine decarboxylase exhibit a suppression of the DCP-induced scratching behavior (*Exp Dermatol* 14, 169–175, 2005) and eczematous lesions (*Arch Dermatol Res* 297, 68–74, 2005). On the other hand, we have previously shown that DCP-induced scratching is attenuated by the suppression of reactive astrocytes that have the capability to enhance GRP-induced scratching behavior (*Nat Med* 21, 927–931, 2015). Spinal GRP–GRPR signaling has been reported to be predominantly involved in histamine-independent itch (*Nature* 448, 700–703, 2007). In addition, our recent unpublished data (manuscript is being revised) found that GRPR KO mice exhibit suppression of DCP-induced scratching (the number of scratching responses: WT, 38739 ± 4505 ; GRPR KO, 19848 ± 1739 (values represent mean \pm S.E.M.); $P=0.0125$, Mann-Whitney U test; $n=9$ –11 mice). Collectively, these data suggest that DCP-induced scratching behavior might involve both histamine-dependent and histamine-independent mechanisms.

REVIEWER COMMENTS

Reviewer #1 (Remarks to the Author):

The authors addressed some concerns. However, several issues raised are either superficially dealt with. Several arguments or claims made throughout the revisions are rather unconvincing. Because acute itch in Nptx2 KO mice is normal, the presence of Nptx2-independent glutamate and neuropeptides for itch, especially the contribution of the GRP-GRPR signaling in acute and chronic itch, and the limitation of their approach should be discussed. In addition, considering that DCP itch is histamine-dependent, the contribution of NMB that is required for histaminergic itch should also be mentioned. Some glutamatergic inputs recorded might also come from spinal NMBR neurons. Overall, the lack of scientific rigor casts doubt on the validity of some claims. I cite two examples below.

1) The authors indicated that it is not feasible to use the intrathecal injection of an antagonist to inhibit DCP-itch because the chronic itch was recorded for 24 h while the drug has a short-term effect. This sounds more like a lame excuse. In the rebuttal letter, the authors also stated "A recent study has demonstrated that intrathecal administration of perampanel, a selective AMPAR antagonist, suppressed the scratching behavior of DCP-treated mice (Biochem Biophys Res Commun 533, 1102–1108, 2020), which supports our findings that NPTX2 strengthens AMPAR-mediated glutamate transmission in GRPR+ itch transmission neurons.". This cited study actually undermines their own argument stated above. Chronic itch behaviors in mice are typically recorded for 30 min or 1 h for analysis (e.g.; Zhao et al; 2013 JCI). Although 24 h was recorded in the study, the effect of the drug can be compared for a short period of time.

2) The authors failed to detect Grp in DRGs using RNAscope, and implied that the sensitivity of RNAscope is good, because spinal Grp mRNA can be detected. However, the fact that Grp is much more abundant in the spinal cord than in DRGs should indicate that spinal Grp may not be an appropriate control.

re: comments of Reviewer 1

The authors addressed some concerns. However, several issues raised are either superficially dealt with. Several arguments or claims made throughout the revisions are rather unconvincing. Because acute itch in Nptx2 KO mice is normal, the presence of Nptx2-independent glutamate and neuropeptides for itch, especially the contribution of the GRP-GRPR signaling in acute and chronic itch, and the limitation of their approach should be discussed. In addition, considering that DCP itch is histamine-dependent, the contribution of NMB that is required for histaminergic itch should also be mentioned. Some glutamatergic inputs recorded might also come from spinal NMBR neurons. Overall, the lack of scientific rigor casts doubt on the validity of some claims. I cite two examples below.

Reply: We thank the reviewer for his/her comments on our manuscript. We agree with the reviewer's suggestion and, in revising the manuscript, have now included discussions regarding the presence of NPTX2-independent glutamate and neuropeptides (e.g., GRP-GRPR signaling) for itch (**lines 283–284, 355–356**) and the contribution of NMB (**lines 359–361**). In addition, as pointed out, considering that acute itch in NPTX2 KO mice is normal, our data provide a new mechanism for chronic itch but not for acute itch (**lines 381–383**), which is a limitation of our approach.

1) The authors indicated that it is not feasible to use the intrathecal injection of an antagonist to inhibit DCP-itch because the chronic itch was recorded for 24 h while the drug has a short-term effect. This sounds more like a lame excuse. In the rebuttal letter, the authors also stated "A recent study has demonstrated that intrathecal administration of perampanel, a selective AMPAR antagonist, suppressed the scratching behavior of DCP-treated mice (Biochem Biophys Res Commun 533, 1102–1108, 2020), which supports our findings that NPTX2 strengthens AMPAR-mediated glutamate transmission in GRPR+ itch transmission neurons." This cited study actually undermines their own argument stated above. Chronic itch behaviors in mice are typically recorded for 30 min or 1 h for analysis (e.g.; Zhao et al; 2013 JCI). Although 24 h was recorded in the study, the effect of the drug can be compared for a short period of time.

Reply: The reviewer mentions that the effect of the GRPR antagonist RC-3095 can be compared for a short period. Unfortunately, it is challenging to perform this approach under our experimental conditions because the number of scratching responses in individual DCP-treated mice is not continuously but intermittently produced (please see **Figure A** below). Thus, the behavioral measurement during a short period (for example, 30 min) will result in a large variation in the number of scratch responses between experimental groups. For this reason, our present and previous studies have monitored scratching responses for 24 h to precisely evaluate chronic itch in this model (e.g., *Nat Med* 21, 927-931, 2015; *J Allergy Clin Immunol* 145, 183-191, 2020). In the paper (BBRC, 2020), they also recorded scratching behaviors for 12 or 24 h and examined the effect of the intrathecal administration of perampanel; however, this drug is a small-molecule compound whose pharmacological kinetics are generally expected to be longer than those of peptide compounds such as the GRPR antagonist RC-3095. In fact, its suppressive effect on scratching behavior is long-lasting (BBRC, 2020). Therefore, because of the lack of pharmacological tools suitable for our experiments

(especially, a GRPR antagonist with a long-lasting effect), we considered that NPTX2/GRP double KO mice would be a better way to determine the role of GRP signaling in DCP-induced chronic itch. However, as in our previous rebuttal letter, it will take more than 1 year to generate NPTX2/GRP double KO mice and complete behavioral experiments. Thus, as described in the response above, we have modified the description (**lines 355–359**) in the Discussion section as follows: *‘it is possible that others, NPTX2-independent glutamate and neuropeptide (e.g., GRP) signaling, may be involved in the residual scratching behavior in DCP-treated NPTX2 KO mice, which requires further investigation using tools (e.g., GRPR-specific antagonists with a long-lasting effect or NPTX2/GRP double KO mice).’*

Figure A
Temporal pattern of scratching behavioral responses in individual DCP-treated mice
 Number of scratching responses (per 1-hour interval) in individual DCP-treated mice (n=8 mice; #1–8) 14 days after the last DCP treatment.

2) *The authors failed to detect Grp in DRGs using RNAscope, and implied that the sensitivity of RNAscope is good, because spinal Grp mRNA can be detected. However, the fact that Grp is much more abundant in the spinal cord than in DRGs should indicate that spinal Grp may not be an appropriate control.*

Reply: To address the reviewer’s concern regarding our RNAscope data, we examined the expression of *Grp* mRNA using a different kit that enables the enhancement of fluorescence signals (RNAscope™ Multiplex Fluorescent Reagent Kit v2). In comparison with the previous data (shown in Figure Aa in our point-by-point responses in the second revision), the fluorescence signals of *Grp* mRNA were more clearly observed in the SDH (**Figure Ba**, below). As another positive control, we have now shown that *Grp* mRNA fluorescence signals were also detected in the suprachiasmatic nucleus (SCN) (**Figure Bb**), where *Grp* has been reported to be expressed (*Science* 355, 1072-1076, 2017), which confirms the specificity and sensitivity of the RNAscope probe used in this experiment. Under such conditions, we observed *Grp* mRNA signals in DRG neurons (**Figure Bc**). However, *Grp* mRNA-positive DRG neurons were rare (one or two neurons per DRG section; **Figure Bc**), and the fluorescence signals were weak (**Figure Bf**; compared with that in the SDH and SCN shown in **Figure Bd,e**). These data are consistent with our qPCR data (please see Figure Ab in our point-by-point responses of the second revision). In addition, similar weak signals of *Grp* mRNA in a few DRG neurons were observed in DRG sections from DCP-treated mice (**Figure Bg**), but these neurons did not have *Nptx2* mRNA signals (**Figure Bg**). From these results, we assume that GRP may not be expressed in NPTX2-positive DRG neurons in the DCP model under our experimental conditions. Nevertheless, several

papers have reported GRP-expressing DRG neurons, whose number is larger than that observed in the present study. The exact reason for this difference in the DRG is unclear, but we consider it might be due to some methodological differences, which is a technical limitation of our study.

Figure B. *Grp* mRNA expression in the SDH, suprachiasmatic nucleus (SCN), and DRG.

a–c, Fluorescence *in situ* hybridization analysis for *Grp* mRNA in sections of the SDH (a), SCN (b), and DRG (c) using RNAScope™ Multiplex Fluorescent Reagent Kit v2. DAPI: 4',6-diamidino-2-phenylindole.

d–f, Highly magnified images of *Grp* mRNA in the dotted square in panel a (SDH: d), b (SCN: e), and c (DRG: f).

g, *Grp* and *Nptx2* mRNA in DRG sections of DCP-treated mice. Results are representative of five experiments.

Scale bars: 150 μ m (a–c) and 30 μ m (d–g).

REVIEWER COMMENTS

Reviewer #1 (Remarks to the Author):

Major:

The conclusion is still misleading as illustrated below. Several directly relevant pieces of literature should be cited.

Line 282-284.

The authors attempted to address the expression of GRP in DRGs and found a few GRP-positive cells. This is an improvement. According to the method provided, the poor sensitivity of RNAscope may be accounted for by the storage of the tissues. It is well known that for the detection of mRNA with low-level expression, the tissue should be freshly prepared and used for RNAscope. This technical limitation should be discussed. It is very likely that GRP expression is expanded to overlap with NPTX2 under chronic itch conditions (PMID: 3809799). Thus, the conclusion regarding an independent glutamatergic pathway under chronic itch conditions seems misleading.

This is supported by the fact that overexpression of NPTX2 failed to induce spontaneous scratching behavior. The role of glutamate is to facilitate itch transmission mediated by neuropeptides rather than function independently as an itch transmitter as the authors indicated (see PMID: 34663954).

Minor:

Line 62: reference 2 is not about GRPR signaling. The original study (PMID: 17653196) and a recent review on GRPR should be cited (PMID: 34663954).

Line 66: one of the first studies describing the increase of GRP/GRPR in chronic itch should be cited PMID 3809799

Line 68: reference 6 has nothing to do with the statement. The first i.t. GRPR antagonist to suppress itch was published and should be cited (PMID: 17653196).

Line 382, "the role of NPTX2" may be restricted to chronic itch." This is not accurate as aforementioned. NPTX2 may well play a role in acute itch. However, it may not be feasible to completely eliminate presynaptic glutamate with existing approaches. This inherent problem has contributed to several conflicting claims in the literature regarding the role of glutamate.

re: comments of Reviewer 1

The conclusion is still misleading as illustrated below. Several directly relevant pieces of literature should be cited.

Line 282-284.

The authors attempted to address the expression of GRP in DRGs and found a few GRP-positive cells. This is an improvement. According to the method provided, the poor sensitivity of RNAscope may be accounted for by the storage of the tissues. It is well known that for the detection of mRNA with low-level expression, the tissue should be freshly prepared and used for RNAscope. This technical limitation should be discussed. It is very likely that GRP expression is expanded to overlap with NPTX2 under chronic itch conditions (PMID: 3809799). Thus, the conclusion regarding an independent glutamatergic pathway under chronic itch conditions seems misleading.

This is supported by the fact that overexpression of NPTX2 failed to induce spontaneous scratching behavior. The role of glutamate is to facilitate itch transmission mediated by neuropeptides rather than function independently as an itch transmitter as the authors indicated (see PMID: 34663954).

Reply: We thank the reviewer's suggestions and comments. While the fluorescence signals of *Grp* mRNA in the SDH and SCN were clearly detected by RNAscope in situ hybridization under our experimental conditions, it is also possible that the detection level of *Grp* mRNA in the DRG (and other tissues) may be further increased by improving some technical points including the storage of the tissues. Although there is currently no actual data indicating the co-expression because of such technical limitations, we cannot completely eliminate the possibilities that NPTX2 and GRP are co-expressed in DRG neurons of models of chronic itch and that the role of glutamate is to facilitate itch transmission mediated by the neuropeptide. A possible involvement of GRP derived from primary afferents in the facilitation of GRPR⁺ neuronal excitation by NPTX2 and glutamate under chronic itch conditions has already been mentioned in the Discussion section of the previous version (line 366–369). According to this discussion, we have now also modified the Supplementary Figure 7 in the revised manuscript.

Line 62: reference 2 is not about GRPR signaling. The original study (PMID: 17653196) and a recent review on GRPR should be cited (PMID: 34663954).

Line 66: one of the first studies describing the increase of GRP/GRPR in chronic itch should be cited PMID 3809799

Line 68: reference 6 has nothing to do with the statement. The first i.t. GRPR antagonist to suppress itch was published and should be cited (PMID: 17653196).

Reply: In revising the manuscript, these papers have been cited.

Line 382, "the role of NPTX2" may be restricted to chronic itch." This is not accurate as aforementioned. NPTX2 may well play a role in acute itch. However, it may not be feasible to completely eliminate presynaptic glutamate with existing approaches. This inherent problem has contributed to several conflicting claims in the literature regarding the role of glutamate.

Reply: Since we cannot completely rule out a possible involvement of NPTX2 in acute itch evoked by other pruritogens, we have now deleted these sentences regarding acute itch in the last paragraph of the Discussion section and also added a description (**line 280–284**).